# Magnetosheath jet evolution as a function of lifetime: Global hybrid-Vlasov simulations compared to MMS observations

Minna Palmroth[1,2], Savvas Raptis[3], Jonas Suni[1], Tomas Karlsson[3], Lucile Turc[1], Andreas Johlander[1], Urs Ganse[1], Yann Pfau-Kempf[1], Xochitl Blanco-Cano[4], Mojtaba Akhavan-Tafti[5], Markus Battarbee[1], Maxime Dubart[1], Maxime Grandin[1], Vertti Tarvus[1], and Adnane Osmane[1]

[1]Department of Physics, University of Helsinki, Helsinki, Finland
[2]Space and Earth Observation Centre, Finnish Meteorological Institute, Helsinki, Finland
[3]KTH Royal Institute of Technology, Stockholm, Sweden
[4]Instituto de Geofisica, Universidad Nacional Autonoma de Mexico, Mexico City, Mexico
[5]Climate and Space Science and Engineering, University of Michigan, Ann Arbor, USA

*Correspondence to:* Minna Palmroth (minna.palmroth@helsinki.fi)

**Abstract.** Magnetosheath jets are regions of high dynamic pressure, which can traverse from the bow shock towards the magnetopause. Recent modelling efforts, limited to a single jet and a single set of upstream conditions, have provided the first estimations about how the jet parameters behave as a function of position within the magnetosheath. Here we expand the earlier results by making the first statistical investigation of the jet dimensions and parameters as a function of their lifetime within the
magnetosheath. To verify the simulation behaviour, we first identify jets from Magnetosphere Multi-Scale (MMS) spacecraft data (6142 in total) and confirm the Vlasiator jet general behaviour using statistics of 924 simulated individual jets. We find that the jets in the simulation are in quantitative agreement with the observations, confirming earlier findings related to jets using Vlasiator. The jet density, dynamic pressure and magnetic field intensity show a sharp jump at the bow shock, which decreases towards the magnetopause. The jets appear compressive and cooler than the magnetosheath at the bow shock, while during
their propagation towards the magnetopause they thermalise. Further, the shape of the jets flatten as they progress through the magnetosheath. They are able to maintain their flow velocity and direction within the magnetosheath flow, and they end up preferentially to the side of the magnetosheath behind the quasi-parallel shock. Finally, we find that Vlasiator jets during low solar wind Alfvén Mach number $M_A$ are shorter in duration, smaller in their extent, and weaker in terms of dynamic pressure and magnetic field intensity as compared to the jets during high $M_A$.

## 1   Introduction

Earth's magnetosheath is a region of turbulent shocked plasma of solar wind origin. The earthward boundary of the magnetosheath is the magnetopause, while the outer boundary is the bow shock, at which the characteristics of the solar wind plasma abruptly change. Much of the magnetosheath global properties can be predicted from gasdynamic or magnetohydrodynamic (MHD) models (e.g., Stahara, 2002; Zastenker et al., 2002; Walsh et al., 2012; Dimmock and Nykyri, 2013), for example,
the magnetosheath plasma is overall denser and hotter as compared to the solar wind. Since the bow shock is a fast magnetosonic shock, the magnetosheath magnetic field is also more intense as compared to the interplanetary magnetic field (IMF)

(Petrinec, 2013). However, magnetosheath physics is inherently kinetic physics, indicating that many magnetosheath characteristics are neglected in the gasdynamical and MHD treatments. For example, mirror mode and Alfvén ion cyclotron waves, excited by temperature anisotropies, are frequently observed in the magnetosheath (Schwartz et al., 1996; Génot et al., 2009; Soucek et al., 2015; Dimmock et al., 2015; Dubart et al., 2020). Further, the alteration of incoming solar wind parameters in the magnetosheath can depart from MHD predictions (Turc et al., 2017), and there may be other kinetic ion-scale features like wave-particle interactions that play a role in the characteristics of the overall magnetosheath (Dimmock et al., 2017). One decisive factor controlling the variability of the magnetosheath in general is the orientation of the IMF with respect to the bow shock surface (e.g. Lucek et al., 2005). The incoming solar wind particles reflect at the bow shock and propagate far upstream (Schwartz et al., 1983). If the IMF is approximately quasi-parallel with the shock normal, the reflected particles cause instabilities and waves in a region called the foreshock upstream of the shock (Eastwood et al., 2005). These foreshock waves advect with the solar wind towards the bow shock and interact with it, and cause increased turbulence and variability within the magnetosheath downstream of the quasi-parallel shock (Němeček et al., 2002; Shevyrev et al., 2007; Dimmock et al., 2014).

Němeček et al. (1998) provided some of the first observations of kinetic structures, which are now called magnetosheath jets, characterised by high plasma velocities and dynamic pressures. Plaschke et al. (2018) present a recent review of the observational features of the jets and their statistical characteristics, including their influence on magnetospheric dynamics. In summary, jets have been associated with the quasi-parallel magnetosheath, suggesting that they originate from interactions between the foreshock and the bow shock (e.g., Archer and Horbury, 2013; Plaschke et al., 2013; Vuorinen et al., 2019). Using a large statistical data base, Plaschke et al. (2013) further demonstrate that the jet durations are around tens of seconds, and are faster, more dense, and have a larger magnetic field intensity relative to the ambient magnetosheath. On the other hand, Plaschke et al. (2018) note that the jets are commonly colder than the surrounding magnetosheath. Despite their localized nature, jets play a role in energy transfer from the solar wind to the magnetosphere, for example in triggering magnetopause reconnection (Hietala et al., 2018) or magnetopause surface waves (Archer et al., 2019).

Modelling studies of the magnetosheath jets so far have concentrated on investigating the properties of single jets appearing in ion-kinetic simulations. Consistent with observational statistics, many modelling studies (e.g., Omidi et al., 2016; Palmroth et al., 2018a) associate the jets to the quasi-parallel magnetosheath, and foreshock–bow shock interactions. Modelling results of single jets within simulations show that the jet characteristics, like the dynamic pressure, velocity and magnetic field intensity appear within the range of observational statistics. The power of modelling tools in investigating the jets is that in simulations single jets can be followed in time, and their characteristics can be investigated as a function of the jet lifetime. Further, in observations, the jet length scales can only be inferred from the time it takes for the jet to traverse over the spacecraft, while in simulations the jet size can be determined as a function of time. Palmroth et al. (2018a) showed the first such study for a single jet that was carefully confirmed to represent a jet based on three different observational criteria. They found that the investigated single jet was an elongated structure, with a length of about 2.6 $R_E$ and width of about 0.5 $R_E$. Further, supporting the idea presented by Karlsson et al. (2015), they found that the jet originated at the bow shock in consequence of a foreshock high dynamic pressure resembling a short, large-amplitude magnetic structure (SLAMS, e.g., Lucek et al., 2002).

In this paper we extend the Palmroth et al. (2018a) investigation to incorporate more than one jet in a single simulation run. We identify a statistical data set of multiple jets in four runs, having a range of solar wind conditions. We concentrate in the role of bow shock normal angle ($\theta_{Bn}$), and solar wind Alfvén Mach number $M_A$, since these parameters are chiefly responsible in influencing the shock properties. Since we deal with simulation results, we first verify the statistical simulation data set by rigorously comparing to a data set collected from the Magnetosphere Multi-Scale (MMS) spacecraft (Burch et al., 2016). After confirming that the simulation data set is in quantitative agreement with the MMS statistics, we investigate the jet properties as a function of time and relative position from the bow shock using the superposed epoch method. Further, we investigate how the jets travel within the magnetosheath and where do they preferentially end up. The paper is organised as follows: First, we introduce the Vlasiator simulation and the runs that are used, along with introducing the MMS data set. We then present an example jet from both data sets and confirm their individual properties, after which we compare the properties statistically. Finally, we move to investigate the jet properties as a function of the lifetime, and end the paper with our discussion and conclusions.

## 2 Methods

### 2.1 Vlasiator

We use Vlasiator (Palmroth et al., 2013; von Alfthan et al., 2014; Palmroth et al., 2018b), which is a global hybrid-Vlasov simulation, propagating protons as distribution functions while electrons are a massless charge-neutralising fluid. Vlasiator solves ion kinetic-scale physics self-consistently by representing the ions in a 3-dimensional velocity space grid (3V). The ion distributions are propagated in time using the Vlasov equation, and the velocity space is coupled to the ordinary space, where electromagnetic fields are solved using Maxwell's equations and complemented by Ohm's law including the Hall term. The power of Vlasiator in comparison to the complementary particle-in-cell method is that the distribution function is noiseless (see, e.g., Kempf et al., 2015). While inherently Vlasiator's ordinary space is 3-dimensional (3D), in this paper we use a 2D simplification due to the computational resources that were available for the set of runs. This makes the total dimensionality of the model here 2D3V.

We present four different runs carried out in the ecliptic $XY$ plane in the Geocentric Solar Ecliptic (GSE) coordinate system. The simulation box for the different runs varies, but is large enough to include the solar wind, foreshock, dayside magnetosheath and parts of the nightside. The resolution is the same for all runs (227 km for the real space, and 30 km s$^{-1}$ for the velocity space), to our knowledge a breakthrough resolution compared to previous hybrid-kinetic simulation studies of jets. We have carefully analysed the minimum requirements for spatial and velocity space resolution (Pfau-Kempf et al., 2018; Dubart et al., 2020) and chosen the run parameters so that the physics of jets is properly described. The inner boundary of the magnetospheric domain is 5 $R_E$ for all runs. The ionospheric boundary is a perfectly conducting circle. The dipole strength has been set to represent the natural dipole strength at Earth, indicating that the modelling results can be given in SI units without scaling. Each run introduces the solar wind conditions in the sunward wall, while copy conditions are applied in the other walls, and in

**Table 1.** Characteristics of the four presented runs. The first column gives the run identifier according to the overall run characteristics: H (L) for high (low) Alfén Mach number $M_A$, respectively, and a number giving the interplanetary magnetic field (IMF) cone angle. IMF vector and intensity, number density $n$, velocity $\mathbf{v}$, IMF cone angle, and $M_A$ are given for all runs. The solar wind temperature for all runs is 0.5 MK. All runs have the same resolution, 227 km for the real space, and 30 kms$^{-1}$ for the velocity space.

|  | IMF [nT] | \|IMF\| | $n$ [cm$^{-3}$] | $\mathbf{v}$ [kms$^{-1}$] | Cone [°] | $M_A$ |
|---|---|---|---|---|---|---|
| HM30 | $(-4.3, 2.5, 0.0)$ | 5 | 1 | $(-750, 0, 0)$ | 30 | 6.9 |
| HM05 | $(-5.0, 0.4, 0.0)$ | 5 | 3.3 | $(-600, 0, 0)$ | 5 | 10 |
| LM30 | $(-8.7, 5.0, 0.0)$ | 10 | 1 | $(-750, 0, 0)$ | 30 | 3.4 |
| LM05 | $(-10.0, 0.9, 0.0)$ | 10 | 3.3 | $(-600, 0, 0)$ | 5 | 5 |

the $Z$ direction periodical boundary conditions are applied. The initial velocity distribution is Maxwellian, which then changes self-consistently when the runs advance.

Table 1 presents the input parameters for the runs in terms of the solar wind and interplanetary magnetic field (IMF). Two of the runs have an IMF cone angle, measured from the Sun-Earth line, of 30°, while the two others have an almost radial IMF. The solar wind Alfvén Mach number $M_A$ within the runs incorporates a spread from 3.4 to 10. The solar wind values have been chosen to accommodate a variability in the $M_A$, while still having the values well within the range of typical values (Winterhalter and Kivelson, 1988), justifying investigations of the foreshock and its interactions with the bow shock. The run set facilitates investigating the magnetosheath jets in terms of the central parameters in shock physics, i.e., the shock normal angle with respect of the IMF direction ($\theta_{Bn}$), and $M_A$. These runs have been used in a variety of investigations of the foreshock and magnetosheath properties (see, e.g., Palmroth et al., 2015; Hoilijoki et al., 2016; Turc et al., 2018; Palmroth et al., 2018a; Turc et al., 2019).

## 2.2 MMS spacecraft and solar wind data

MMS was launched on March 12, 2015. Since then it has provided measurements using a comprehensive suite of plasma instruments (Burch et al., 2016). In this work, we use the Fast Plasma Investigation (FPI) for plasma moment data, taken in the Fast Survey mode (with one measurement per 4.5 s) (Pollock et al., 2016) until May 2019. For magnetic field measurements, we use Survey mode data (8 s$^{-1}$) from the Fluxgate magnetometer (FGM, Russell et al., 2016). We exclusively use data from MMS1, since due to the small separation of the satellites, multi-spacecraft methods are not necessary in our mostly statistical analysis. Due to the close spacing of the satellites, all satellites provide comparable measurements. The solar wind parameters used to normalise MMS data are retrieved from the OMNI data base (King and Papitashvili, 2005).

## 3 Example jet: Modelling and Observations

In earlier jet studies several criteria for defining and identifying magnetosheath jets have been used. For example, Plaschke et al. (2013) compare magnetosheath dynamic pressure (evaluated using the $x$ component of the ion velocity) to the solar wind dynamic pressure, and demand that the former should be at least 25% of the solar wind pressure. Archer and Horbury (2013) used only magnetosheath data and defined a jet as an enhancement of the dynamic pressure as compared to the ambient magnetosheath value, and defined a "dynamic pressure enhancement" by demanding that

$$\frac{P_{dyn}}{\langle P_{dyn}\rangle} \geq 2 \tag{1}$$

where the dynamic pressure is given by $P_{dyn} = \rho v^2$ using the plasma mass density $\rho$ and velocity $v$. The angular brackets in Eq. (1) denote a temporal average, which Archer and Horbury (2013) implemented as a running average with a 20 min window. The benefit of this Archer criterion is that it eliminates the need of solar wind data, which are not always available, or which may be fluctuating strongly, making it difficult to decide which data should be used in the normalisation. In this paper we apply the Archer criterion, however, in Vlasiator we average over three minutes and keep the 20 minutes for the MMS data. The three-minute running average centred at the time of interest in the Vlasiator data was already deemed accurate by Palmroth et al. (2018a), who compared three different jet criteria within the Vlasiator simulation.

For the MMS statistics, we initially find the time intervals in which MMS resides inside the magnetosheath region. This is carried out using thresholds for the ion density, temperature, velocity and flux that are set manually. In addition, we require that the magnetosheath interval lasts at least 15 minutes, to avoid possible influence of the bow shock or magnetopause. After determining the magnetosheath intervals, we use the *in situ* measurements of ion velocity and density to compute the dynamic pressure, and apply the Archer criterion in Eq. (1) using a 20-minute average time window. We require that the jet is identified at least 15 minutes away from the magnetopause and the bow shock so that the time average is not affected. Furthermore, we impose a minimum time separation between jets ensuring that sequential jet-like measurements, which appear less than a minute apart, but which are likely part of the same structure, are treated as a single event. MMS data has been used to investigate jets e.g., by Raptis et al. (2020). To match the MMS statistics with the solar wind parameters, we time-shifted and averaged OMNI data for an ideal match to the jets observation at the magnetosheath. This was done by taking an average of 20 (1-min resolution) data points from OMNI, starting 15 minutes before the jet observation time and up to 5 minutes after the jet observation time. This unequal averaging was done because the average time from the bow shock to MMS location is ~5 minutes as discussed in Raptis et al. (2020). As a result, by taking under consideration the time lag, we effectively take a $\pm 10$-minute window around the associated solar wind measurements for the jets. With this technique we remove the extremely varying solar wind conditions, and take care of the time shift required from the bow shock to magnetosheath.

The identification of jets in Vlasiator data is performed as follows. Figures 1a and 1b show a snapshot from Run LM30 at time $t = 410$ s. The full run is presented as a Supplementary movie S1. Colour-coding shows the dynamic pressure. While the large-scale flow pattern in the magnetosheath in Vlasiator is as expected, diverging flow from the Sun-Earth line, the distribution of the velocity magnitude and the density is more complex due to kinetic processes arising at the quasi-parallel shock. Thus,

the dynamic pressure within the magnetosheath illustrates remnants of the ULF wave fluctuations at the bow shock, shown as large-scale structures, or "stripes" that can be seen within the dynamic pressure (see their formation and propagation from the bow shock in the Supplementary movie S1). The red line shows a fit to the bow shock location, where this position is defined as the location where the heating of the core population is larger than three times the solar wind temperature, similarly as in Battarbee et al. (2020a). Identifying the magnetopause position in these 2D3V runs is complicated (Palmroth et al., 2018a), and thus in this paper we concentrate more in regions near the bow shock. The magenta contour shows as a reference the regions where the Plaschke et al. (2013) criterion is in force using the 25% threshold. Figure 1a shows that the Plaschke et al. (2013) criterion identifies large regions particularly towards the flanks as jets. Since these areas are likely those at which the magnetosheath flow starts to accelerate towards the nightside, and hence not likely jets, in this paper we use the Archer and Horbury (2013) criterion for identification. We additionally require that the jets are smaller than 4500 cells, which corresponds to a surface area amounting roughly to 6 $R_E$ by 6 $R_E$. Further, to limit small ripples from being identified as jets, we require that the jets are larger than 2 cells (0.05 $R_E$ by 0.05 $R_E$). Hence, Figs. 1a-b show a black contour delineating jets that are smaller than 4500 cells and larger than 2 cells using the Archer and Horbury (2013) criterion. Figures 1a-b further show red and white dots, which are best visible in Fig. 1b. The red dots give the average centre position of the jets defined simply as the average position of all positions within the contour delineating the jet. The white dots indicate the locations of the largest velocity within the jet. Figures 1a-b show many jets, most of them appearing near the bow shock surface. Most noticeably, Figs. 1a-b show a jet approximately at [$X,Y$]=[11.8, $-1$] $R_E$. This jet is investigated in detail in Fig. 2.

Figure 2 shows Vlasiator virtual spacecraft and MMS data from the position shown by the stars in Fig. 1b (orange for Vlasiator and green for MMS). This jet was chosen because its position in Vlasiator and in MMS are at a closest possible proximity in all four runs such that the velocity distribution is available within the Vlasiator jet. Although the Vlasiator simulation solves the velocity space everywhere, it is not written to file everywhere due to restrictions in disk space. The MMS data are gathered on 3 Dec, 2015, during which the solar wind conditions are the following: Velocity is around 410 $\mathrm{kms}^{-1}$, density about 3.8 particles per cubic centimetre, the magnetic field vector about [$-4$, 3, $-2$] nT, and $M_A$ is $\sim$7. Table 1 presents the solar wind conditions for Run LM30: Velocity is 750 $\mathrm{kms}^{-1}$, density 1 particle per cubic centimetre, magnetic field vector [$-8.7$, 5, 0] nT, and $M_A$ is 3.4. Despite the small difference in the location at which the data are given and the discrepancies in the solar wind conditions, the Vlasiator and the MMS jets show a good temporal correspondence. The pressure increases in a similar manner from outside of the jet to within the jet (the black vertical dashed line indicates the peak pressure within the jet). The velocity and the magnetic field show some discrepancies, such as slower flows and more variable magnetic fields at MMS, reflecting the differences in the solar wind conditions as well as the possible different relative positions within the magnetosheath. The energy spectrogram in Vlasiator is a directional integral of phase-space as a function of energy and time, similar to the method introduced in Jarvinen et al. (2018). This spectrogram shows similar particle energies as the MMS data, slightly larger in Vlasiator because of the larger solar wind velocity. Similarly, the differences in the density and temperature can be understood in terms of the differences in the solar wind parameters. The main purpose of Fig. 2 is to show that the jet identified in Vlasiator has a similar behaviour in time as the jet in the MMS data in terms of the dynamic pressure, the main criterion to identify the jet.

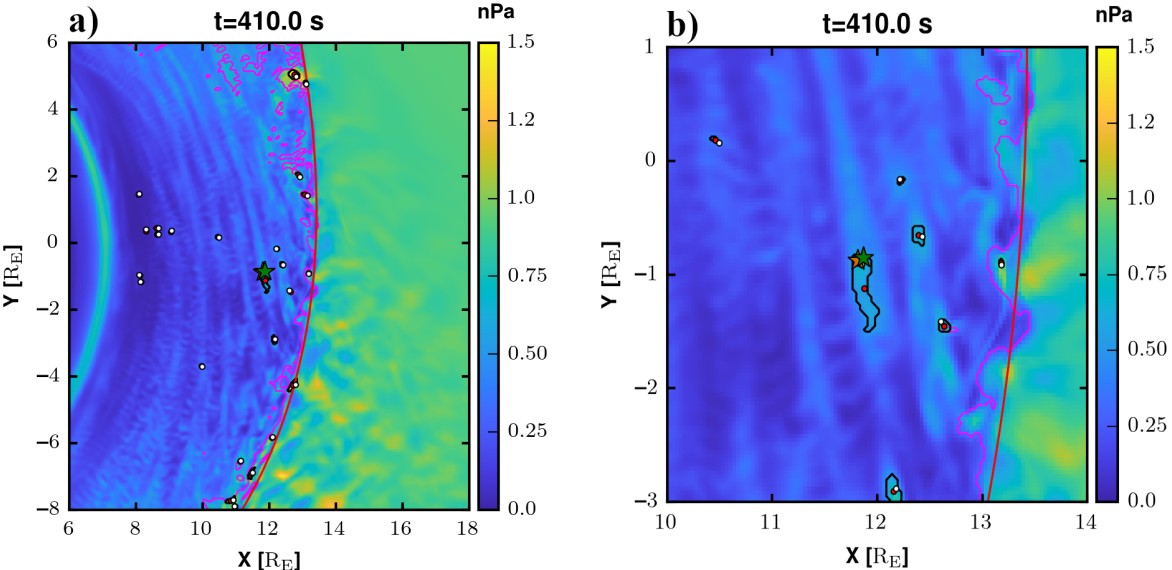

**Figure 1.** a) A zoom to the magnetosheath in Vlasiator Run LM30 at time 410 s from the start of the simulation. Colour coding shows the dynamic pressure in units of nPa. The magenta contour shows as a reference the regions where the Plaschke et al. (2013) criterion is at force using the 25% threshold. The black contours indicate jets identified with the Archer and Horbury (2013) criterion. The red line is an approximate fit of the bow shock location. Red dots indicate the average centre position of the jet, while the white dots show the location of largest velocity. The star refers to the location at which the virtual satellite and MMS data are taken and shown in Fig. 2. b) Same as panel a) but a further zoom to show the jets in more detail. The two star positions indicate where data are shown in Fig. 2 (orange for Vlasiator and green for MMS).

.

## 4 Statistical comparison of the jets

We now present a statistical comparison of the jets identified in the four Vlasiator runs, to the ones identified from the MMS data. This section first introduces the automatic jet identification and tracking method developed to identify the jets in Vlasiator and follow their evolution, after which it briefly introduces the MMS data set. Finally, we present the statistical jet characteristics using both Vlasiator and MMS.

### 4.1 Vlasiator and MMS jet data sets

In Vlasiator data, we apply the Archer and Horbury (2013) criterion in Eq. (1) using the three minute running average centred at the jet, as in Palmroth et al. (2018a). Since each simulation run spans hundreds of seconds, and we save files twice a second, we give each jet an identity. This ensures that we are able to investigate the jets as a function of their lifetime. The jet identity is given as follows: At any given time step, the jet criterion in Eq. (1) delineates simulation cells as described in Sect. 3. The search is carried out in $[X;Y]$ boxes of $[6, 18; -8, 6]$ $R_E$ for the HM30 and LM30 runs, and $[6, 18; -6, 6]$ $R_E$ for HM05 and

LM05 runs. Search areas were chosen to focus on the bow shock nose, with additional space in the $-Y$ direction for the $30°$ cone angle runs to capture more of the foreshock, while omitting most of the flanks and regions of accelerating magnetosheath flows. In the next time step, the jet is regarded as the same as during the previous time step, if at least 50% of the cells are the same as those that belonged to the jet during the previous time step. In cases where one predecessor jet has two or more

successors (i.e. splintering of a jet), the cells of the successor with the largest number of cells are considered to belong to the mother jet, and the remaining successor jets are considered to be new jets. Two jets can also coalesce, in which case the one which came into existence later, disappears, i.e., the dynamic pressure decreases so much that the Archer and Horbury (2013) is not met anymore. We require that the jet lifetime is at least 5 seconds. We further define that if a jet disappears and does not reappear in 10 seconds, it is considered dead. This is to mimic the MMS data set requirement in which jets less than one

minute apart of each other belong to the same structure.

The supplementary movie S1 shows the temporal evolution of the jets, and one can follow the path of the red dots to see where the jet started and where it ended. The animation shows also that sometimes the red dots appear in the middle of the magnetosheath, and do not originate at the bow shock. These features are identified as jets because the local conditions fulfil the jet criteria. In the literature, the term magnetosheath jets is reserved for features that are associated to the bow shock - foreshock

interactions (e.g., Archer and Horbury, 2013), indicating that features appearing far from the bow shock are not the jets that are meant by the term. Nevertheless, we adopt an inclusive strategy and include all events fulfilling the jet criterion. This is to increase similarity to the observational statistics, as none of the statistical studies of jets using spacecraft measurements have an opportunity to confirm the place of their origin, or whether the magnetosheath flow conditions altered due to e.g., kinetic mode waves such that the local parameters fulfil the jet criteria. Based on Supplementary movie S1, we suspect that this may occur in

many statistical data sets. Further, based on the Supplementary movie S1 we also limit to smaller jets and avoid large regions near flanks, because we suspect that the jet criterion may falsely identify regions where the flow becomes super-Alfvénic as jets.

Applying the jet criterion and the other restrictions described above to the identification routine, we found a total of 924 jets from the four runs, whose $XY$ GSE positions are visible in Fig. 3 as purple and orange dots representing low and high

$M_A$ runs, respectively. The number of jets in the different runs are also presented in Table 2. Since the runs are carried out with different solar wind conditions, the position of the magnetopause and bow shock is not the same between the runs, and therefore the average positions of the magnetopause (Shue et al., 1998) and the bow shock (Merka et al., 2005) do not represent the reality for an individual jet observation. The jets appear at different distances along the $X$ axis in each run, reflecting the solar wind conditions and the consequent magnetosheath position in each run. They are evenly distributed in terms of their $Y$

position within the magnetosheath. The largest number of jets are found in Run LM30, which has the lowest $M_A$ and a $30°$ IMF cone angle. This run was carried out longer than the other runs, indicating that the Run LM30 is over-emphasised in the statistical results.

The MMS data set consists of jets fulfilling the Archer criterion using the 20-minute averaging. To facilitate comparison with Vlasiator, we impose a stability criterion to the solar wind. This is to increase confidence that the jets included in the statistics

contain stable solar wind conditions and therefore, can be meaningfully compared with Vlasiator results, which are obtained

**Table 2.** Jets in the four runs listed in Table 1. The first (second) column is the start (stop) time of the simulation when the jet search began (ended). The third column gives the number of jets found in each run.

|  | Jet search start [s] | Jet search stop [s] | Number of jets |
|---|---|---|---|
| Run HM30 | 290 | 419.5 | 144 |
| Run HM05 | 290 | 589.5 | 293 |
| Run LM30 | 290 | 669.5 | 368 |
| Run LM05 | 290 | 439.5 | 119 |

with synthetic stable solar wind. This stability criterion requires that the maximum standard deviation of the magnetic field rotation angle 15 minutes before and 5 minutes after the jet is smaller than $45°$. As a result, the MMS dataset contains 6142 jets from beginning of mission data to end of May 2019. This set is further divided into low and high $M_A$ ($M_A < 6$, and $M_A > 6$, respectively), resulting in 577 and 5533 jets, respectively. Figure 3 shows the high (low) $M_A$ jets found from the MMS data as purple (light blue) crosses, respectively.

## 4.2 Statistical properties of the jets: Vlasiator and MMS results

Before we present the statistical properties of the Vlasiator and MMS jets, we summarise the differences of the observation criteria in Vlasiator and in MMS. Figure 4 illustrates the lifetime evolution of an imaginary jet starting at the bow shock (marked as BS in Fig. 4), and ending up near the magnetopause (MP). The black line delineates the jet at 5 different times of its life on its journey from the bow shock to the magnetopause. The red dot represents the geometric centre position of the jet. In the following statistical data set, the Vlasiator jets can be depicted at any time during their life and their properties can be obtained at a given time or as a function of the jet lifetime. In contrast, the MMS crosses the jet at a random time at a random position. To facilitate the comparison between Vlasiator and MMS, in the following statistics we present the Vlasiator data in two different manners. First, we describe the Vlasiator jet properties at time "VLMax", i.e., at the time when the jet is at its largest size, and at the position where the parameter has its maximum value within the jet (see Fig. 4). However, as it is not known whether the MMS crosses the jet when it is at its largest size, or where the crossing takes place within the jet, we also present the Vlasiator data at a time "VLRand", i.e., at the position where the parameter is at its maximum, at a random time of the jet evolution. This manner is adopted to other parameters except for temperatures, which are averaged from within the jet, not taken at the maximum value due to a large variation of the temperature. Figure 4 also presents the MMS crossing of the jet in blue, taking place at a random position through the jet at a random time during the jet lifetime. Further, since our run set does not represent all solar wind values detected in the MMS observational data set, we make the statistical comparison with respect to the solar wind in the following manner. First, in Figures 5-6 we use normalisation to the solar wind values as units both in the Vlasiator and in the MMS data. Second, in Figure 6 we show the results using values from which the background has been subtracted. While variation in the observations will certainly depend on the solar wind conditions, by subtracting the background conditions and normalising to the solar wind we minimise this effect.

Figure 5 presents the statistical properties of the Vlasiator and MMS jets. The first two columns refer to Vlasiator times VLMax and VLRand as illustrated in Fig. 4, while the third column gives the MMS jet statistics. From top to bottom, each row depicts a histogram of the jet extent, density, speed, dynamic pressure, magnetic field magnitude, and temperature, respectively. The parameter "Extent" in Vlasiator is the size of the jet in $R_E$ in the $X$ direction, and it is compared to the distance which the MMS traverses within the jet as determined by the identifying criterion. All other parameters except the extent and temperature are normalised to the respective solar wind values. Taking into account that the simulation results represent four solar wind conditions, while the MMS data are gathered during a larger variety of conditions, Figure 5 shows an overall agreement especially between the Vlasiator jets at random times and the MMS jets. Possibly due to the discrepancy in determining the jet extent, the MMS jets appear slightly larger than the Vlasiator jets, and they span a broader range of scales. The second row describes the maximum density inside the jets, indicating that the median density of the Vlasiator (MMS) jets at random times is 4.9 (5.9) times the solar wind density. The shapes of the Vlasiator and MMS jet maximum velocity and maximum dynamic pressure distributions are in good general agreement, and the median values are in quantitative agreement. The maximum magnetic field shows a narrower distribution in Vlasiator as compared to the distribution using the MMS data, and the MMS jets are also more intense in terms of the magnetic field (the median value of the maximum magnetic field inside the jet is 6.2 times larger than the IMF, while in Vlasiator during random time it is about 3.1 times larger than the IMF). The last row shows the temperature distributions of the jets. The Vlasiator jets are about three times hotter than in the MMS observations both in the parallel and in the perpendicular direction. There is also a separation in the Vlasiator jets, such that the perpendicular temperature is larger than the parallel temperature. The MMS distribution shows no such separation, and the median temperature both in parallel and perpendicular direction is about 3 MK.

Figure 6 shows a similar histogram setup as in Fig. 5. Instead of the value within the jet, we now present a histogram made by subtracting the ambient magnetosheath value from the jet value. To do this, we define a thin shell of two simulation cells outside the jet area, take the average of this shell, and subtract it from the jet maximum value. For temperature (the first row showing a histogram of ΔT) we subtract the shell average temperature from the average temperature within the jet. The normalised histograms are also separated such that blue (red) colour indicates jets during low (high) $M_A$ in the solar wind. The threshold between high and low $M_A$ is 6 in both Vlasiator and MMS. Figure 6 again shows an excellent agreement between Vlasiator and MMS data both in terms of distribution shapes as well as quantitatively. Overall, no large differences appear between the low and high $M_A$ distributions other than in the magnetic field intensity. In the last row of Fig. 6, we note that during high $M_A$, the magnetic field intensity differences between the jet and the magnetosheath are consistently larger, both in Vlasiator and in MMS. In other words, during high $M_A$, the jet magnetic field increases relative to the magnetic field within the magnetosheath, as compared to jets during low $M_A$.

## 5   Jet properties: Vlasiator results

According to the Figures 5 and 6, the Vlasiator jet statistics are in a good quantitative correspondence with the MMS statistics. Therefore we next present Vlasiator results of jet characteristics using the power of the modelling tool describing e.g., the

time evolution of the jet. Figure 7 shows Vlasiator histograms of the jet lifetime, tangential size (at its largest), and size ratio, defined as the jet radial size divided by the tangential size at the time when the jets are at their largest extent. The radial size is computed as the difference of the maximum and minimum points of the jet in the radial direction from the centre of the Earth. The tangential size is the jet total area divided by the radial size. All histograms show the normalised number of jets. Figure 7a shows that most of the jets in all runs are short-lived with a median lifetime less than 11 seconds. However, in all runs there are jets which can last over 50 seconds. Interestingly, there is a larger share of short-lived jets during low $M_A$ (green and orange histograms in Fig. 7a) than during high $M_A$. The tangential size (Fig. 7b) is around 0.1 $R_E$ in all runs, with little variability between the driving conditions. The size ratio (Fig. 7c) is mostly slightly larger than 1, indicating jets of $\sim$0.1 by 0.1 $R_E$, with the radial size perhaps slightly larger. Again, only small variability is detected in the size ratio histograms between runs.

Figure 8 presents a modified superposed epoch study of the Vlasiator jets in all runs. By "modified", we mean that we do not use time in the epoch axis, but convert the time series at the centre of the jet to a position profile given at the $X$ axis. Depicted are the jet extent, tangential size, and size ratio, as a function of the jet position relative to the bow shock. The bow shock position is defined to be where the heating of the core population is larger than three times the solar wind temperature, similarly as in Battarbee et al. (2020a). The epoch $X$ axis is defined from the difference of the bow shock and jet $X$ coordinates, indicating that 0 means bow shock, and larger numbers indicate deeper regions towards the magnetopause. The zero epoch position is chosen when the jet is at the bow shock. Figure 8 shows that some jets appear upstream before the 0 epoch position. This is because jets are identified up to 0.5 $R_E$ sunward of the polynomially fitted bow shock so as not to exclude jets in regions where the shock is locally sunward of the fit. The different colours give the modified superposed epoch average for the different runs, with green and orange (purple and black) representing the low (high) $M_A$ runs, respectively. Grey shows standard deviations subtracted and added to the average, taken from all jets.

Figure 8 shows a consistent difference between the high and low $M_A$ runs, as in the high $M_A$ runs the jets show a larger extent, and a larger tangential size as compared to the low $M_A$ runs. The size ratio is smallest for the Run LM30 (which has a 30° IMF cone angle with low $M_A$), while for the other runs the size ratio does not change much according to the driving conditions. Although in Fig. 7c the size ratio median is slightly larger than 1, according to Fig. 8c, the size ratio can be over 2 at maximum, indicating that the radial size prevails. A clear feature shown in Fig. 8b-c is that jets are flattening as they propagate deeper into the magnetosheath: the tangential size increases and the size ratio decreases from the bow shock towards the magnetopause. To verify this behaviour, we identified a particularly strong jet and followed its size parameters from the bow shock to come to this same conclusion (not shown).

Figure 9 shows a similar superposed epoch study as a function of the jet position relative to the bow shock as in Fig. 8, however, now we present the difference between the jet and the surrounding magnetosheath environment. For example, the top panel presenting the difference in density is calculated by subtracting the average magnetosheath density (in a two-cell shell around the jet) from the maximum density value inside the jet. Therefore, each epoch curve represents how much the jet value increases or decreases from the neighboring magnetosheath as a function of position from the bow shock. Figure 9 shows again consistently that the jets during high solar wind $M_A$ are denser, faster, and they have a more intense dynamic pressure and magnetic field relative to the surrounding magnetosheath, compared to the jets during low $M_A$. The density, velocity,

dynamic pressure and magnetic field intensity epoch curves all behave similarly: after the bow shock these parameters show an initial maximum first, and towards the magnetopause they decrease from the maximum. The density, velocity, dynamic pressure and magnetic field intensity are positive all the way, indicating that the jet itself shows larger values than those within the magnetosheath. The temperatures behave differently, as in the vicinity of the shock the temperature differences are

generally lower and negative, and increase towards the magnetopause. This means that the jets start cooler than the surrounding magnetosheath and thermalise as they progress through the magnetosheath. Overall, there is considerably more variation in the temperature. The 30° cone angle run with high Mach number (black curve) shows the coolest values within the jet relative to the surrounding magnetosheath.

To show how an individual, particularly strong jet behaves in time, we show in Fig. 10 one jet identified from the Run HM05

(purple curves in Figs. 7-9) as a function of the simulation time rather than position from the bow shock. The properties are taken from the centre of the jet at each time step. The general behaviour is the same as in Fig. 9, however, the individual jet shows interesting features that are smeared out in the overall statistics. The jet density increases fast after the bow shock indicating initial compression. The dynamic pressure and magnetic field intensity show also a stronger increase after the bow shock, suggesting compressive behaviour. Towards the end of its lifetime, the density, velocity, dynamic pressure and

magnetic field intensity generally decrease. Both temperatures are cooler than the surroundings when the jet was born, while the structure thermalises towards the end of its life. Based on Figure 10 the jet ceases to exist through diffusion rather than disintegration, because towards the end of its lifetime there are no steep gradients within the jet parameters (without counting the final seconds).

Next, we investigate how the jet moves through the magnetosheath relative to the overall flow; again a feature that can

only be studied using a global model. Figure 11 shows the jet velocity deflection away from the Sun-Earth line relative to the background magnetosheath flow. The plot construction is similar to Figs. 8-9 as a modified superposed epoch is presented, and the epoch $X$ axis is defined from the difference of the bow shock and jet $X$ coordinates. Figure 11a shows the magnitude of the velocity deflection relative to the background and it is constructed in the following manner: $|v| = \sqrt{v_x^2 + v_y^2}$, where $v_x$ and $v_y$ are the components of the jet velocity in the $X$ and $Y$ directions, respectively. The magnetosheath velocity $< |v| >$

is the average magnetosheath velocity at each respective jet position during the duration of the run. In effect, this average magnetosheath velocity thus represents the average flow of the magnetosheath within the position where the jet is at any given time during its progression within the magnetosheath. Figure 11a shows that for all other runs except Run LM30, $|v| - < |v| >$ is positive (indicating that the jet progresses faster than the background magnetosheath flow, and that during the progression towards the magnetosheath the jet velocity slows down towards the magnetosheath average. For the Run LM30, curiously,

the jets progress faster than the magnetosheath flow near the bow shock, but then $|v| - < |v| >$ becomes negative on average, meaning that the jets travel slower compared to the magnetosheath when they get further from the bow shock.

Figure 11b shows the deflection angle, where $\theta = \arctan(|v_y|/ - v_x)$ is the deflection angle away from the Sun-Earth line, and $< \theta >$ is the average of the deflection angle of the magnetosheath within the position where the jet is at any given time during its progression within the magnetosheath throughout the run. Therefore Figure 11b illustrates how much on average the

jet velocity is deflected from the Sun-Earth line relative to the background flow pattern. Figure 11b shows that for all other runs

except the Run LM30, the deflection angle is negative, indicating that the jets on average travel more along the Sun-Earth line than along the magnetosheath flow. In other words, the average flow pattern within the jet position is more towards the flanks relative to the jet flow direction. However, in the run LM30, $\theta - <\theta>$ is positive, indicating that these structures flow more towards the flanks than the average flow direction with the jet position.

5     Finally, in Figure 12 we investigate where the jets end up within the magnetosheath. Figure 12a shows the jet end coordinate in $Y$, while Figure 12b shows the jet end angle in the $XY$ plane, and black (purple) lines give the results for the $30°$ ($5°$) cone angle, grouping both the low and high solar wind $M_A$ runs. Figure 12 thus illustrates the difference in event occurrence patterns regarding location behind the quasi-parallel and quasi-perpendicular bow shock. Figure 12 clearly shows that the jets tend to end up in the dawn flank of the magnetosheath especially during the $30°$ cone angle runs, that is, to the side of the magnetosheath which is behind the quasi-parallel bow shock. This same tendency can be seen also in the almost radial runs, while it is slightly smaller in magnitude.

## 6   Discussion and conclusions

In this paper we have rigorously compared magnetosheath jets identified from the Vlasiator simulation in four runs with jets identified from MMS observations. We confirm that the Vlasiator jet properties are statistically in an excellent quantitative agreement with the observational MMS statistics. Further, we note that individual jets from Vlasiator and MMS suggest quantitative agreement in the dynamic pressure, even though the jets are identified during different solar wind conditions. After noting the quantitative agreement, we show Vlasiator results of the statistical behaviour of the jets as a function of lifetime and relative position from the bow shock. We find that overall, Vlasiator jets during low solar wind $M_A$ are shorter in duration, and smaller in their extent as compared to the jets during high $M_A$. The shape of the jets changes as they progress through the magnetosheath. Further, we confirm the findings of Plaschke et al. (2013) in that the jets are faster, denser, and have a higher dynamic pressure and more intense magnetic field as compared to the ambient magnetosheath near the bow shock. They are also cooler relative to the magnetosheath. Towards the magnetopause, the jet characteristics start to be more magnetosheath-like. In addition, we find that the largest deviations from the ambient magnetosheath values occur just after the bow shock, while towards the magnetopause the parameters start to approach the magnetosheath values. The jets appear first compressive and cooler than the magnetosheath, while during their propagation they thermalise. The jets are able to maintain their flow through the magnetosheath and they preferentially end up to the side of the magnetosheath which is behind the quasi-parallel shock.

     Two assumptions have been made in Vlasiator to reach these conclusions about magnetosheath jets, i.e., we carry out the simulations in 2 spatial dimensions (2D) and assume that electrons are massless charge-neutralising fluid. Both simplifications are made due to huge computational demands that are already in place with the 2D3V approach that requires a supercomputer with a large run-time memory. In fact, no 3D global kinetic or hybrid-kinetic simulations of jets currently exist to which these results could be compared to. The bow shock - magnetosheath interactions are rather accurately reproduced by 2D kinetic simulations as can be seen from the spacecraft comparison shown in this paper and from a large number of previous papers

(e.g., Blanco-Cano et al., 2006; Karimabadi et al., 2014). The largest caveat of the 2D3V approach is that the position of the magnetopause is difficult to determine, which is why we avoid making conclusions near the magnetopause.

As for neglecting the electron dynamics in a hybrid scheme, first we note that the kinetic pressure of the electrons downstream of the Earth's bow shock is smaller by about a factor of 10 with respect to the ion pressure. The fact that the observational jet dimensions are between fluid and ion kinetic scales indicates that an ion kinetic model neglecting kinetic electrons should be sufficient to investigate jet sizes. To understand the electromagnetic effects due to the absence of kinetic electrons, one must consult kinetic simulations including both ions and electrons, which on the other hand have to limit the simulation volumes due to computational restrictions. Voitcu and Echim (2016, 2018) investigate the propagation of jet-like features through a plasma having a transverse magnetic field. They find that a polarisation electric field forms inside the jet-like feature, contributing to the forward propagation of the jet-like feature across the transverse background magnetic field. This process is neglected from our simulation in the absence of kinetic electrons. It is difficult to speculate what the additional polarisation electric field would cause in our simulations, especially as our simulations have a radial to Parker spiral conditions. In the radial runs (HM05 and LM05) the magnetic field is highly turbulent at the subsolar magnetosheath, and in the HM30 and LM30 runs it is a bit less turbulent but definitely not purely transverse (not shown). Therefore this effect could be studied perhaps using the new eVlasiator (Battarbee et al., 2020b) with electron capabilities to investigate this aspect in the future.

According to Plaschke et al. (2016), the jet scale sizes may convey information about how the jets came into existence, and therefore it is important to characterise the jet size distributions. The observational statistics have so far characterised the jet sizes in two directions, parallel and perpendicular to the direction of propagation, while Vlasiator results characterise the jets in radial and tangential directions. Another point which influences the direct comparison is that in Vlasiator the size is determined as the area delineated by the jet criterion contour, while in observational statistics it is determined essentially from the time that it takes for the jet to traverse past the spacecraft. Further, to determine the jet perpendicular size, multi-spacecraft methods need to be employed, and therefore not many statistics can be found where both dimensions are estimated. However, Archer et al. (2012) find that the perpendicular (parallel) size is between 0.2 and 0.5 $R_E$ (1 $R_E$), while we, using the same criterion find the mean tangential size about 0.1 $R_E$ and that the radial size is dominating over the tangential size. Plaschke et al. (2016) find that the sizes are 1.34 $R_E$ and 0.71 $R_E$ in perpendicular and parallel direction, respectively, while Plaschke et al. (2020) report scale sizes of 0.12 $R_E$ and 0.15 $R_E$. Our results are hence in excellent agreement with the reported literature regardless of the identification criteria..

The evolution of the jet size is very difficult to investigate using spacecraft observations, and therefore to our knowledge our results give the first estimations on how the jets evolve as they progress through the magnetosheath. Our results show that the jet shape seems to be flattening, and the tangential size increases while the size ratio decreases from the bow shock towards the magnetopause. The size ratio decreases for all runs, and especially so in Run LM30, which is the run with the lowest $M_A$. In this run (given with the orange line in Fig. 8) the jets seem to flatten within the first $R_E$ within the magnetosheath, while for other runs the flattening occurs further downstream. Figure 9c indicates that the dynamic pressure and magnetic field intensity for the LM30 run jets is smallest compared to the other runs throughout their journey from the bow shock towards the magnetopause. Hence, the rapid flattening of the LM30 jets, together with the fact that they do not maintain their velocity

against the magnetosheath flow, is consistent with the notion that these jets appear to be "weakest" in the statistics, and they slow down most rapidly in the magnetosheath. This suggests that the strength of the jets is related to the strength of the shock.

Continuing with the differences organised by the $M_A$, we find that the extent of the jet, giving the jet instantaneous size in the $X$ direction, is larger for the high $M_A$ runs as compared to the low $M_A$ runs. The jets associated with low $M_A$ are also shorter in duration as compared to the jets during high $M_A$. This interesting feature may be related to the nature of the foreshock during the different conditions. Turc et al. (2019) demonstrated that the foreshock wave field is different during different $M_A$, and that for low $M_A$ the wavelengths and the transverse extents of the foreshock monochromatic waves were much smaller as compared to foreshock waves during high $M_A$. The characteristics of the foreshock may be related to the origin of the jets at least in two ways: First, Burgess (e.g. 1995) suggests that the bow shock ripples are modulated by the foreshock waves. Therefore, if the foreshock waves are small, the bow shock ripples, suggested to be the origin of the jets (Hietala et al., 2009), are also small and the jet becomes small. In this scenario, the perpendicular size of the jet is coupled to the perpendicular scale of the ULF wave, and the parallel size of the jet is coupled to the parallel wavelength of the foreshock ULF wave. Second, however, it is also possible that the smaller foreshock ULF waves during low $M_A$ do not steepen to form strong compressional structures (like SLAMS) which would easily travel through the bow shock to form the jets as suggested by Karlsson et al. (2015) and Palmroth et al. (2018a).

We also perform a modified superposed epoch study of the jet parameter evolution from the bow shock towards the magnetopause, and find that for all jets the jet density, dynamic pressure and magnetic field intensity increase fast after the bow shock but then decrease gradually towards the magnetopause. Initially both parallel and perpendicular jet temperatures are cooler than the magnetosheath but then tend to reach values corresponding to the adjacent magnetosheath, as the jet propagates deeper into the magnetosheath. This behaviour is clearer for the high $M_A$ runs. The fact that the jets thermalise towards the magnetopause, and they are able to maintain their velocity through the background flow direction, indicate that the jets have individual dynamics, independent from the background magnetosheath. Perhaps they could be thought to be like the injection of a bubble of cold air into hotter air, which eventually mixes with the surroundings. We note that the differences highlighted by the superposed epoch analysis are also visible in the MMS histograms (Fig. 6o), where the median value and the distribution of the magnetic field are following the trend shown in Figs. 9-10. This evolution of the size and the characteristics of the jet suggests a possible initial compression in the density and magnetic field, followed by a possible expansion or diffusion on the way towards the magnetopause. The results indicate that the jets need to push against the ambient magnetosheath, losing their initial momentum at the bow shock.

We find that as the jets traverse through the magnetosheath, they maintain their speed within the magnetosheath flow and propagate generally along the Sun-Earth line as compared to the surrounding plasma. This behaviour is in good correspondence with a number of statistical studies (e.g., Archer and Horbury, 2013; Plaschke et al., 2013). In fact, Plaschke et al. (2013) conclude that the median value of the deflection angle as compared to the magnetosheath flow is 28.6°, while Archer and Horbury (2013) report smaller deflection angles, which are of the order of a few degrees. Our results are in good quantitative correspondence with the Archer and Horbury (2013) observations. We also find that the jets end up preferentially to the side of the magnetosheath which is behind the quasi-parallel shock.

Finally, it is interesting to contemplate why the jet temperatures during their lifetime are so variable (as depicted by the large deviations from the average values shown in grey in Fig. 9e-f), while the other parameters are more organised. As noted, the jets are cooler than the ambient magnetosheath especially at the bow shock but thermalise towards the magnetopause. This might suggest that the jets are formed of solar wind plasma, supporting the idea that they are SLAMS traversing through the magnetosheath (Karlsson et al., 2015; Palmroth et al., 2018a). If the jets would be generated within the magnetosheath, temperatures could be higher as the plasma is already processed at the bow shock crossing. However, the temperature profiles are very variable and there are also individual jets which are generally hotter than the ambient magnetosheath. In part, this might be due to the overall larger temperatures of the Vlasiator magnetosheath, rooting to the fast solar wind velocity. The other possibility might be that there could be several origins for the jets.

*Author contributions.* All co-authors participated in this investigation. JS was mainly responsible in preparing the Vlasiator data and making the Vlasiator figures, while SR did the same for MMS. MP outlined the paper and wrote it. All co-authors contributed in discussions and commented the manuscript.

*Competing interests.* The authors declare that they have no conflict of interest.

*Acknowledgements.* This paper was outlined and drafted in the Third International Vlasiator Science Hackathon held in Helsinki, 19-23 August 2019. The Hackathon was funded by the European Research Council grant 682068 - PRESTISSIMO. We acknowledge The European Research Council for Starting grant 200141-QuESpace, with which Vlasiator (http://helsinki.fi/vlasiator) was developed, and Consolidator grant 682068-PRESTISSIMO awarded to further develop Vlasiator and use it for scientific investigations. We gratefully also acknowledge the Finnish Centre of Excellence in Research of Sustainable Space (Academy of Finland grant number 312351), and Academy of Finland grant numbers 309937, and 328893. The CSC − IT Center for Science in Finland and the PRACE Tier-0 supercomputer infrastructure in HLRS/Stuttgart (grant numbers PRACE-2012061111 and PRACE-2014112573) are acknowledged as they made these results possible. LT acknowledges Marie Sklodowska-Curie grant 704681 and the Academy of Finland grant 322544. SR and TK acknowledge Swedish National Space Board grant 90/17.

Code and data availability: Vlasiator is distributed in http://github.com/fmihpc/vlasiator. This address contains links to the analysator software used to produce the figures. The runs described here take several terabytes of disk space, and they are stored at the CSC − IT Center for Science and at the University of Helsinki. Vlasiator uses an in-house developed data structure (.vlsv format), which is compatible with the VisIt visualisation software (https://wci.llnl.gov/simulation/computer-codes/visit) with a plugin available at the above address. Data presented in this paper can be accessed by following the data policy on the Vlasiator web site. The solar wind data are retrieved from OMNIweb (https://omniweb.gsfc.nasa.gov)

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

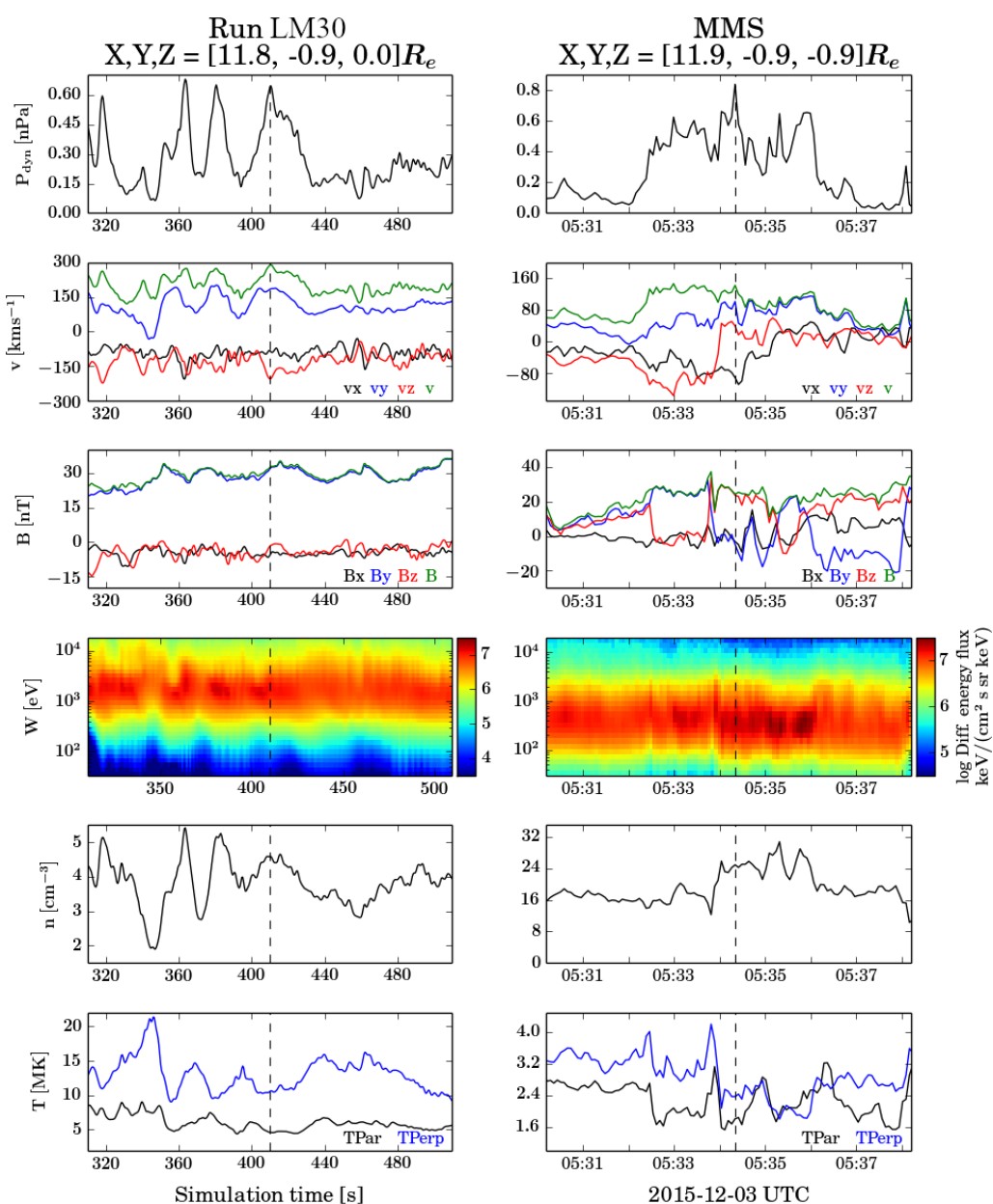

**Figure 2.** (left) Virtual spacecraft and (right) MMS data from the position indicated by a star in Fig. 1a-b. The position is given in $GSE$ coordinates in Earth radius ($R_E$), added on top of the panels. The positions are close to each other but not identical, because the Vlasiator velocity distributions are not saved everywhere. From top to bottom in both Vlasiator and MMS we show the dynamic pressure, three components and the magnitude of the velocity, three components and the magnitude of the magnetic field, the ion energy-time spectrogram, density, and temperature.

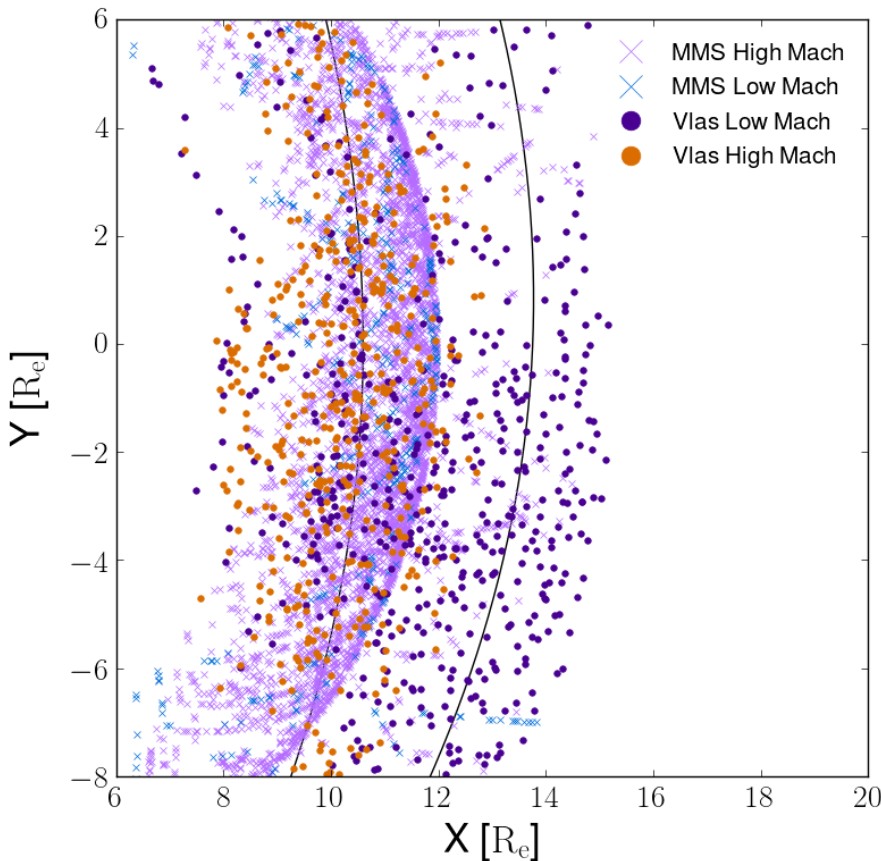

**Figure 3.** Jets position in the $XY$ plane in GSE coordinate system. The jets identified in Vlasiator four runs described in Table 1 are shown as purple (orange) dots, which represent low (high) $M_A$ from Runs LM30 & LM05 (HM30 & HM05), respectively. The total number of jets is 924. Only one position during their lifetime is depicted, this is when the jet was at its largest extent. In purple and light blue crosses we give the position of the high and low $M_A$ MMS jets, respectively. A total of 6142 jets are found from the MMS data. The number of low and high $M_A$ categories is 577 and 5533, respectively, for the MMS, using the same $M_A$ criterion as for Vlasiator. The approximate magnetopause and bow shock positions are given by black solid lines.

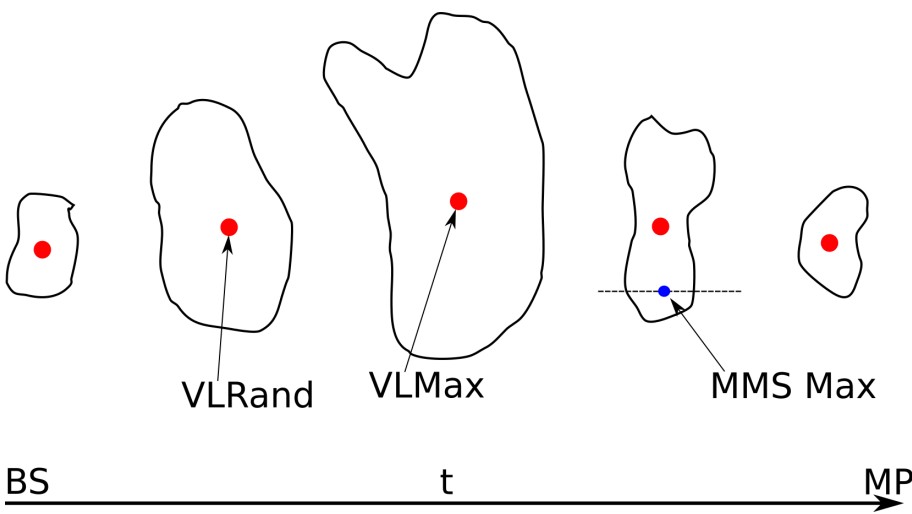

**Figure 4.** Illustration of the differences in detecting jets from Vlasiator and from MMS data (see text for details). The Vlasiator statistical data are given at two different times: the time of the maximum size of the jet ("VLMax"), and at a random time ("VLRand"). In both cases the data is retrieved from the position of the largest value, except for temperatures. Temperatures are averaged over all positions within the jet. The time or place of the MMS jet crossing is not known relative to the jet lifetime, therefore MMS data is given when MMS is observing the jet, i.e., at time "MMS Max" illustrated in blue.

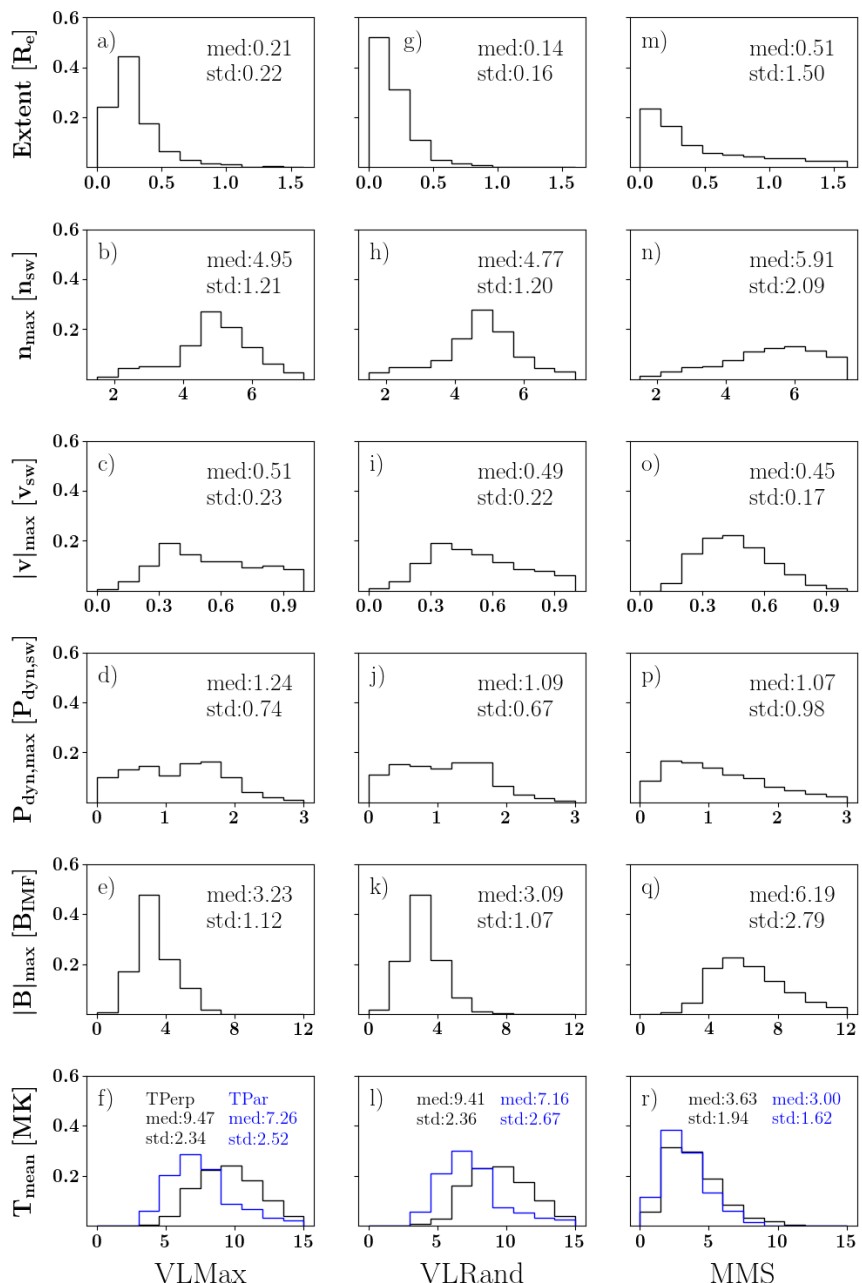

**Figure 5.** Histograms of Vlasiator (two first columns) and MMS jets (third column) at times and positions specified in Fig. 4. From top to bottom we present histograms of the jet total extent, maximum density, maximum velocity, maximum dynamic pressure, maximum magnetic field intensity, and maximum temperature. The extent is given in Earth radii, and the temperature in megakelvins, however, all other parameters are given as normalised to the respective solar wind and IMF values. The bottom row gives both the parallel and perpendicular temperatures in blue and black, respectively. The top right corner of each panel gives the median and standard deviation of the respective histogram.

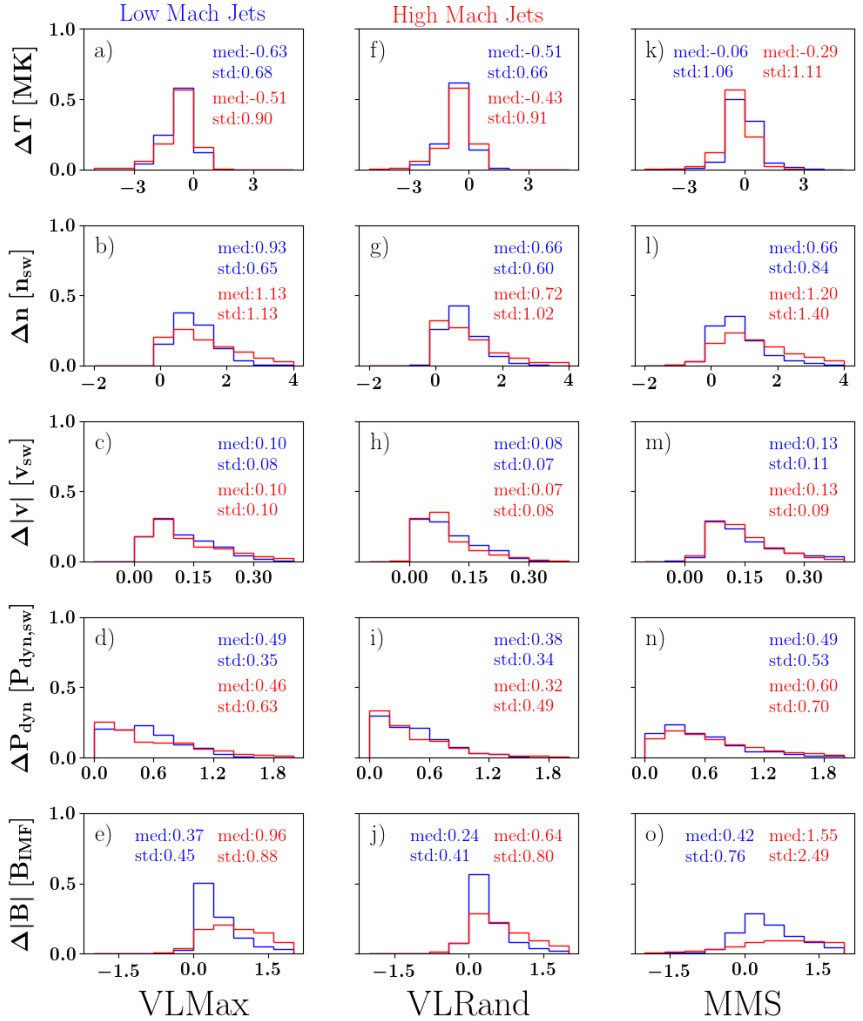

**Figure 6.** Histograms of Vlasiator and MMS jets in a similar setup as in Fig. 5. Now, the histograms present a difference between the value within the jet and the value outside the jet. From top to bottom we present the difference of the jet and magnetosheath temperature, density, velocity, dynamic pressure, and magnetic field intensity, all again normalised to the solar wind values. Red (blue) indicates the data set having higher (lower) $M_A$.

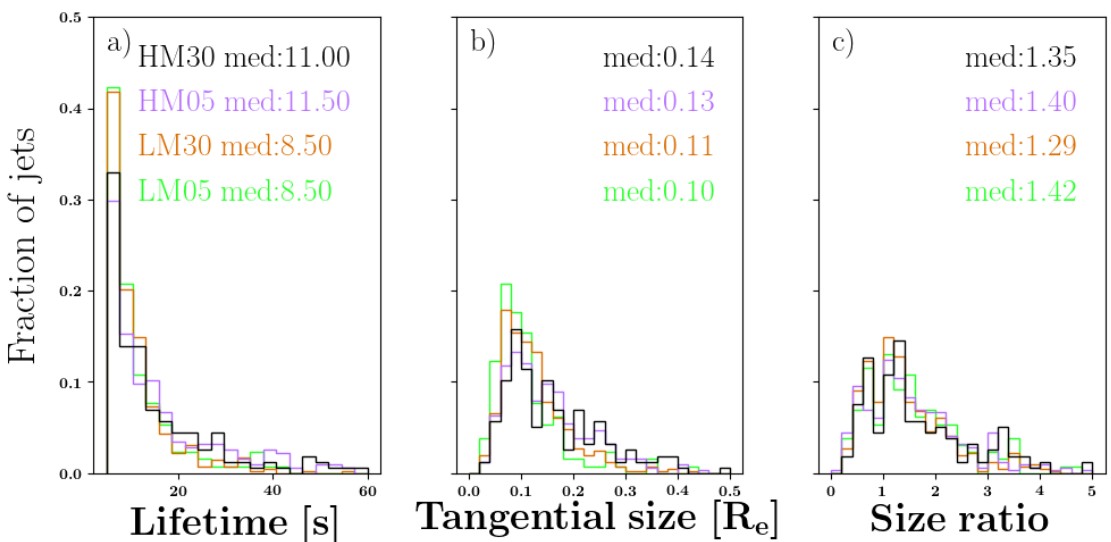

**Figure 7.** Histograms of Vlasiator jets illustrating a) the jet lifetime, b) the jet tangential size, and c) the jet size ratio, defined as the radial size divided by the tangential size. The colours refer to the different runs, with the run identifier (see Table 1): Orange and green (black and purple) are the low (high) $M_A$ runs, respectively.

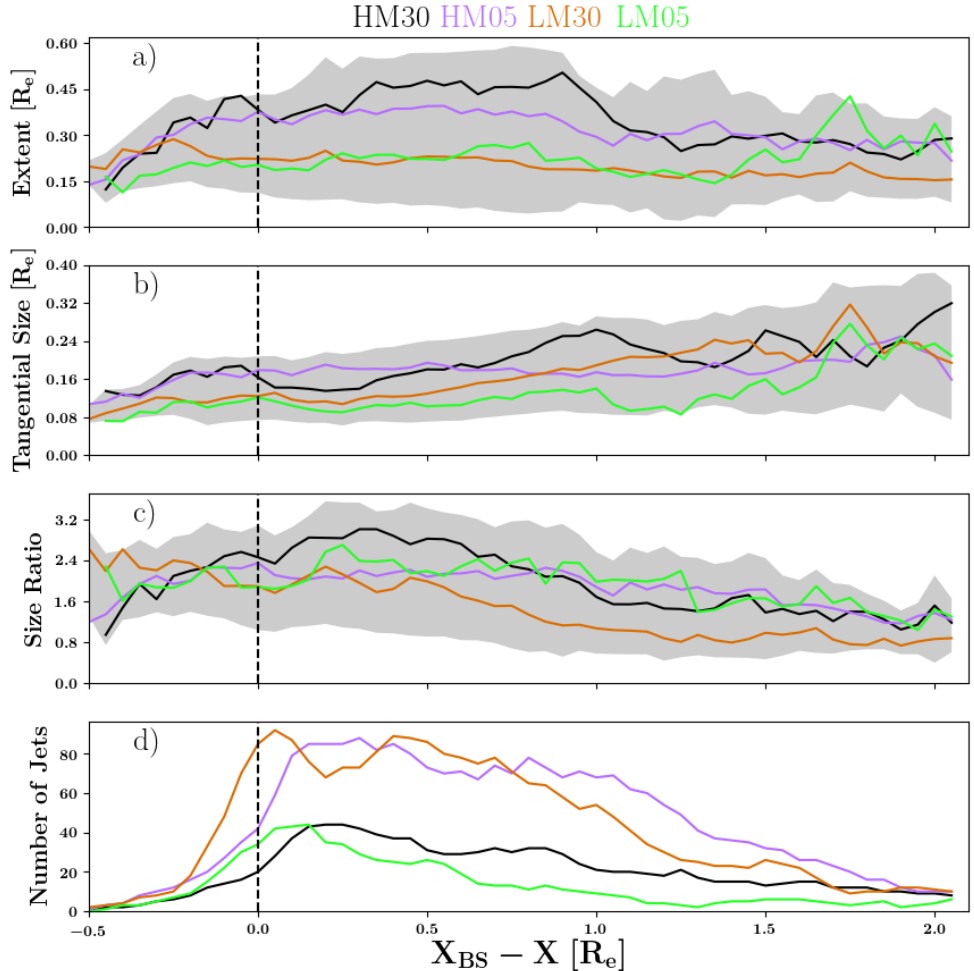

**Figure 8.** Superposed epoch study of the a) extent, b) tangential size, c) size ratio (defined as in Fig. 7c) and d) number of jets as a function of the jet average center position (red dot in Fig. 1a-b) relative to the bow shock. The different colours refer to the superposed epoch averages from the different runs, identified with a identifier (see Table 1): Orange and green (black and purple) are the low (high) $M_A$ runs, respectively. Grey shows standard deviations subtracted and added to the average, taken from all jets.

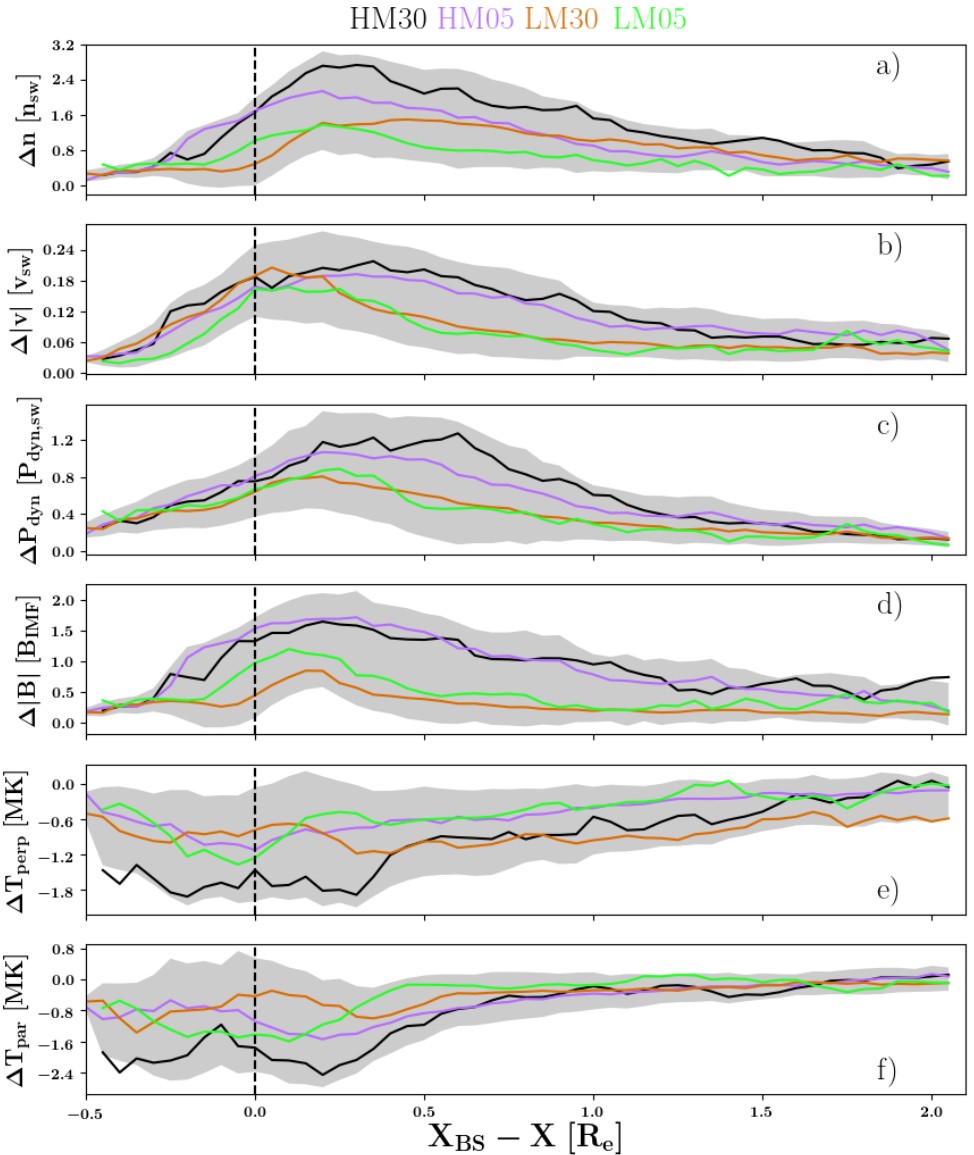

**Figure 9.** Superposed epoch study of the difference between the jet and magnetosheath a) density, b) velocity, c) dynamic pressure, d) magnetic field intensity, e) perpendicular temperature, and f) parallel temperature as a function of jet position relative to the bow shock. All other parameters are normalised to the solar wind, while the temperature data are given in megakelvins. The different colours refer to the superposed epoch averages from the different runs, identified with a identifier (see Table 1): Orange and green (black and purple) are the low (high) $M_A$ runs, respectively. Grey shows standard deviations subtracted and added to the average, taken from all jets.

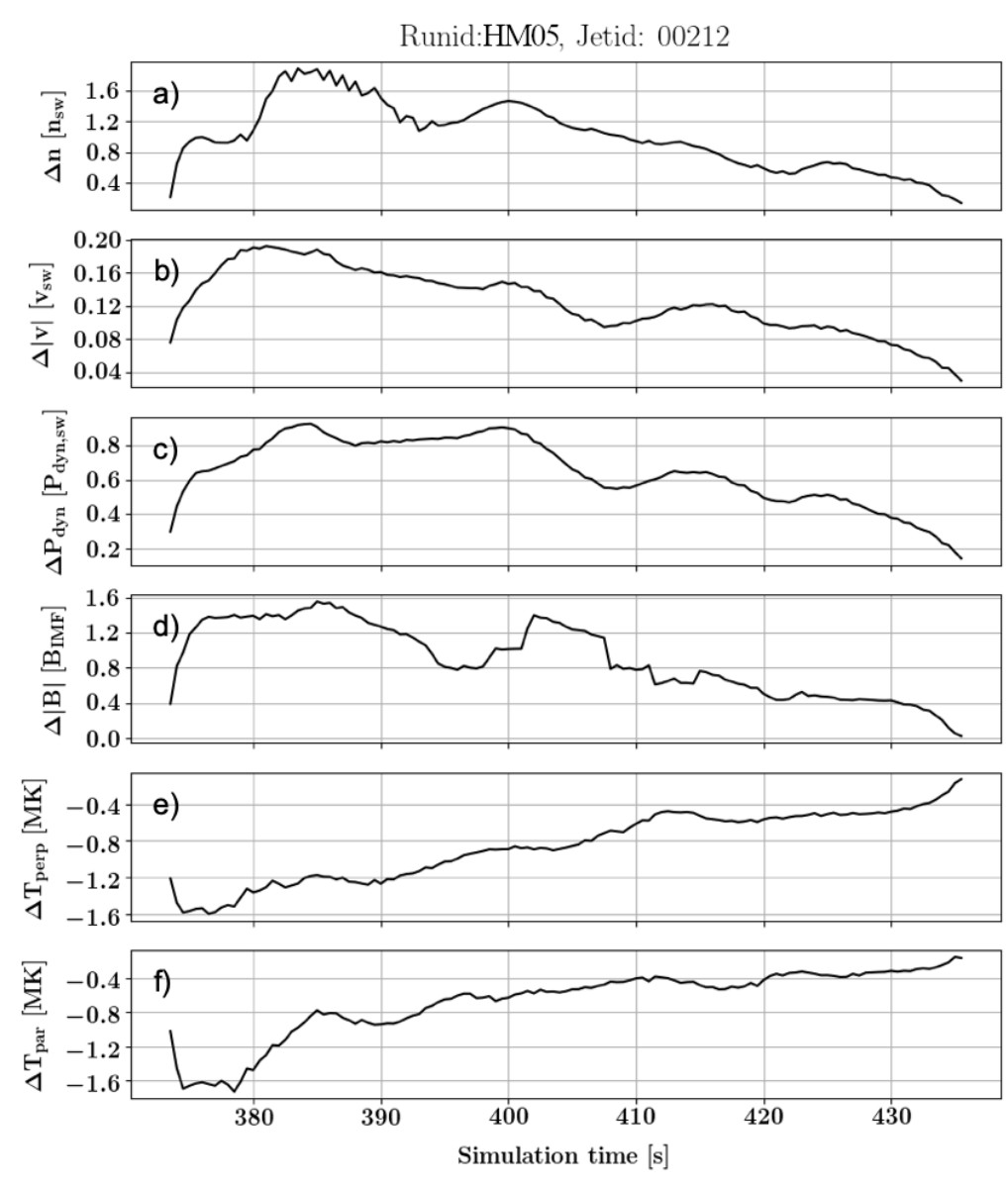

**Figure 10.** The difference between the jet and magnetosheath a) density, b) velocity, c) dynamic pressure, d) magnetic field intensity, e) perpendicular temperature, and f) parallel temperature as a function of jet lifetime in the simulation for a chosen particularly strong jet in the Run HM05 (purple curves in Figs. 7-9). All other parameters are normalised to the solar wind, while the temperature data are given in megakelvins.

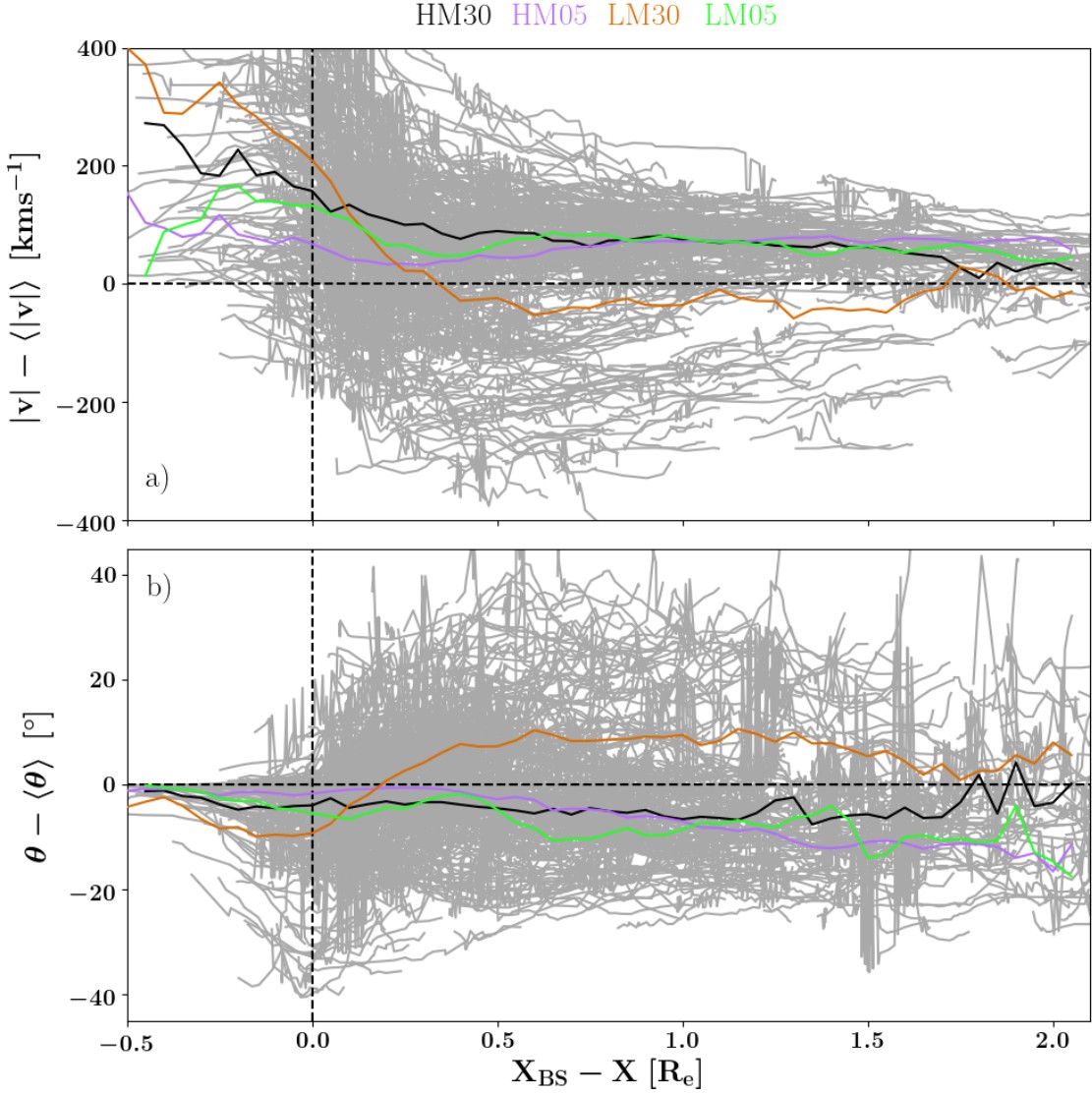

**Figure 11.** Jet velocity deflection away from the Sun-Earth line relative to the background magnetosheath velocity, in terms of a) magnitude of velocity deflection, and b) angle of the velocity deflection. See text for details on the definition of the depicted parameters. The grey lines indicate all events, while different colours refer to the superposed epoch averages from the different runs, identified with a identifier (see Table 1): Orange and green (black and purple) are the low (high) $M_A$ runs, respectively.

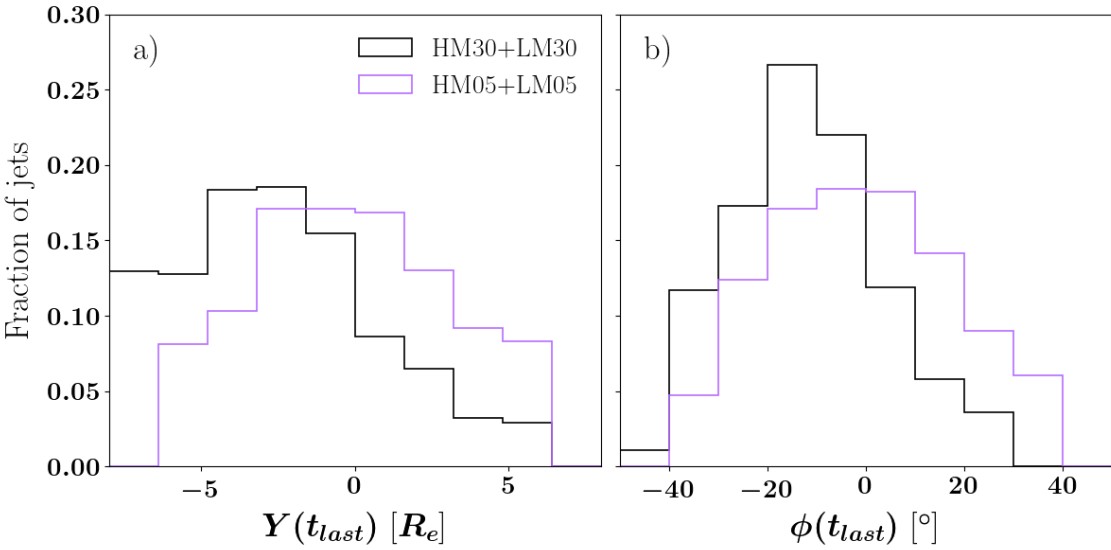

**Figure 12.** Jet end position within the magnetosheath in terms of a) the $Y$ coordinate and b) $\Phi = \arctan(Y/X)$, where $[X, Y]$ are the last coordinates of the jet when it was still detected. Black line gives the jets during $30°$ cone angle, while the purple lines are the results for the $5°$ cone angle.