# Peer review of "Magnetosheath jet evolution as a function of lifetime: Global hybrid-Vlasov simulations compared to MMS observations"

_Annales Geophysicae, 2020_

## Referee Comment (RC1) · David Sibeck (Referee) · 3 Aug 2020

This is an enjoyable paper to read. It has limited objectives, but it fulfills them admirably. The paper first demonstrates that the Vlasiator model successfully reproduces those features of magnetosheath jets that can be observed by individual spacecraft. It then goes on to predict those aspects that cannot be observed by individual spacecraft, namely dimensions and evolution of the jets, as well as their occurrence patterns for different solar wind conditions.

I have three scientific requests for the authors and a few editorial comments. But this paper is basically ready for publication as it stands.

[Figure]

My first scientific request is for the authors to add one parameter to their study. They already consider densities, speeds, temperatures, and magnetic fields within the jets, but I request they add another parameter: deflection of flow away from the sun-earth line. I think it would be interesting to know if the jets maintain flow along this line as they approach the magnetoppause or are deflected just like the background magnetosheath plasma flow.

My second scientific request is that the authors tell whether or not there is a difference in event occurrence patterns in their simulation results regarding location behind the quasi-parallel and quasi-perpendicular bow shock. I am expecting their would be more events in the dawn than the dusk magnetosheath for the spiral IMF orientations they simulate. Can the authors please add some words about that?

My third scientific request is that the authors tell what the vertical pressure stripes are in the magnetosheath. They are so prominent in Figure 1. What is causing them?

Line 6, page 1. Using a statistics –> using statistics Line 21, Page 5. At force –> in force The stars in Figure 1 are too small. Lines 5-7 on Page 10. Can the authors state a typical dimension at this point? Line 22 on Page 10. No need for 'RE' Lines 25-26. Page 10. Can the authors state that the jets appear to be disintegrating or dying by diffusion here?

---

## Referee Comment (RC2) · Anonymous Referee #2 · 17 Aug 2020

The paper by Palmroth et al. discusses results obtained by four runs of a 2D3V hybrid Vlasov code, where the ions are treated kinetically and the electrons as a massless fluid. The manuscript focuses on results of simulations from four sets of initial conditions assumed for the upstream solar wind.

There are two major limitations of these simulations: the kinetic treatment is truncated (the kinetic physics of electrons is disregarded) and the plasma macroscopic dynamics is reduced to two spatial dimensions, in the equatorial plane. Can the authors discuss these limitations and demonstrate that the numerical simulations based on such a truncated description of plasma dynamics still captures accurately the physics of

jets as observed in the magnetosheath (see, e.g., the analysis of the jet anatomy by Karlsson et al, 2018)?

The usefulness of the comparison between a statistical analysis of simulations and MMS observations of jets (collected over almost four years) is not clear to me. The authors state that MMS data serve to "verify the simulation behavior", thus, they aim to validate the hybrid Vlasov simulations with MMS data. However, there is no indication on the upstream solar wind conditions for the 6142 jets detected by MMS1. On the other hand, the statistical ensemble of 924 jets observed in hybrid Vlasov simulations is generated by only four sets of solar wind parameters (magnetic field, density, bulk velocity, cone angle and Mach number), as shown in Table 1. Therefore I am wondering how could one relate, compare and validate simulations of the magnetosheath jets as resulting from these four sets of solar wind parameters with the observations of magnetosheath jets corresponding to 6142 MMS events, and therefore for a much larger set of solar wind conditions ? The authors do show that MMS observations for one jet (figure 2) correspond to a solar wind velocity "around 410 kms$-1$, density about 3.8 particles per cubic centimetre, the magnetic field vector about $[-4, 3, -2]$ nT, and MA is 7" relatively close to the initial conditions assumed for the simulations reported in case LM30. However, even for this case there are significant differences in terms of jet characteristics, as outlined by the authors: "The velocity and the magnetic field show some discrepancies, such as slower flows and more variable magnetic fields at MMS, reflecting the differences in the solar wind conditions as well as the different relative positions within the magnetosheath. [...] Similarly, the differences in the density and temperature can be understood in terms of the differences in the solar wind parameters". Although in Figure 5 is suggested a normalization with respect to solar wind parameters, it is not clear where the solar wind parameters are derived from in case of MMS data.

The supplementary video material raises additional questions. The magnetosheath can be roughly identified in this movie as the spatial region between (an imaginary)

[Figure]

magnetopause (as in Figure 3 of the manuscript – the Shue model) and the red line (the bow-shock). (The authors point out the magnetopause cannot be easily identified in their simulations). It looks like the dynamical pressure irregularities do not follow the general streaming or the magnetosheath flow as they seem to be predominantly convected in the duskward direction, which is puzzling. From the general gasdynamics of a supersonic flow around an obstacle one would expect a stagnation point in the nose region and azymuthal flows, duskward and dawnward, at the two flanks. Also, one notes that some jets (identified by red dots) apparently occur spontaneously at significant distances from the bow-shock, in the magnetosheath, where they spend some time and then disappear ("die") (e.g. the sequence between t=551 and t=557 seconds, roughly at X=12, Y=1). I am not sure this behavior really captures the dynamics of magnetosheath jets which are believed to be structures born at the bow-shock (e.g., Karlsson et al., 2018). When watching the movie in the supplementary material I am wondering if some of the structures considered in this study are not in fact local magnetosheath fluctuations (some of them related perhaps to the inherent numerical simulation noise and/or truncated physics) and not jets, as they are defined in the literature.

The statistical analysis of simulations results is interesting. However, given the complexity of this type of numerical simulations, the authors should give more evidence on how the results depend on the numerical setup and noise. I assume such tests were extensively performed and the authors should discuss if and how the results depend on the spatial and velocity resolution, as well as the time step assumed for the simulations. This is important for understanding the results of the statistical analysis.

The jet size distribution, as revealed by Figures 7 and 8, indicates that the structures studied with the hybrid Vlasov simulations are rather small scale and short lived, with most probable sizes of the order of hundred of kilometers and most probable lifetimes of the order of seconds. While lifetimes were not yet investigated with in-situ data, scale sizes derived from observations (e.g. Plaschke et al, 2016) are orders of magnitude larger, as noted by the authors themselves. This difference leads me back to the

remark I made in the previous paragraph: are the structures studied in this report magnetosheath jets or local non-homogeneities downstream the bow-shock, and satisfying the selection criteria based on the dynamic pressure excess (Eq. 1) ?

Finally, I have a question for the authors. This manuscript, and many others dedicated to magnetosheath jets, presents statistics of various parameters of jets in order to describe their properties. Thus, it is implicitly recognized that these jets are plasma structures to which we can assign a shape, a size, a bulk velocity, a temperature, a lifetime. My question for the authors is which is, in their opinion, the physical process that grants the jets their individual identity, allowing them to have an individual dynamics, apparently different/independent than/from the background magnetosheath.

I see an issue with the availability of simulations and MMS data on which the findings of this manuscript are based. I am wondering if there is compliance with Annales Geophsycae policy on data availability. Indeed, the authors indicate Acknowledgments that "Data presented in this paper can be accessed by following the data policy on the Vlasiator web site." On the Vlasiator website I found the following statement regarding simulations data availability: "All data requests and other support questions should be addressed to the PI. The PI-team decides about the time and place in which the peer-reviewed data becomes public." I think the authors should demonstrate the simulations and experimental data are available from a public data repository as requested by Annales Geophysicae data policy (https://www.annales-geophysicae.net/about/data_policy.html), I quote : "It is particularly important that data and other information underpinning the research findings are "findable, accessible, interoperable, and reusable" (FAIR) not only for humans but also for machines. Therefore, Copernicus Publications requests depositing data that correspond to journal articles in reliable (public) data repositories, assigning digital object identifiers, and properly citing data sets as individual contributions. [...] A data citation in a publication resembles a bibliographic citation and needs to be included in the publication's reference list. [...] In addition, data sets, model code, video supplements, video abstracts,

International Geo Sample Numbers, and other digital assets should be linked to the article through DOIs in the assets tab."

Minor remark: Section 2.2. consists on a single paragraph. I suggest the authors expand it and include a description of the MMS data used in their study or integrate the paragraph in the previous section.

---

## Author Comment (AC1) · 1 Sep 2020

Dear David,

Thank you for your thorough review of our paper. We have addressed all of your very good points which we think will significantly improve the quality of the paper. We go through the questions in the below separate document, where your original comments are in italics.

Best regards, Minna

Please also note the supplement to this comment:
https://angeo.copernicus.org/preprints/angeo-2020-49/angeo-2020-49-AC1-supplement.pdf

**Supplement:**

Dear David,

Thank you for your thorough review of our paper. We have addressed all of your very good points which we think will significantly improve the quality of the paper. Below, we go through the points in detail.

*"My first scientific request is for the authors to add one parameter to their study. They already consider densities, speeds, temperatures, and magnetic fields within the jets, but I request they add another parameter: deflection of flow away from the sun-earth line. I think it would be interesting to know if the jets maintain flow along this line as they approach the magnetopause or are deflected just like the background magnetosheath plasma flow. "*

This is an excellent suggestion. It can be deduced rather easily: We can look at the relation V_yz/V_x within the jet as average, median and maximum value. Another possibility is to look at the path of the jet and compare this to the path of a random plasma parcel within the background magnetosheath flow. We shall add this characteristic to the paper.

*"My second scientific request is that the authors tell whether or not there is a difference in event occurrence patterns in their simulation results regarding location behind the quasi-parallel and quasi-perpendicular bow shock. I am expecting there would be more events in the dawn than the dusk magnetosheath for the spiral IMF orientations they simulate. Can the authors please add some words about that?"*

This is very interesting question, to which we can at least partly answer in the paper: In the revised version, we can compare the normalised number of events in the 30-degree cone angle runs between dawn and dusk, concentrating to the final position of the jet. Indeed, you are right, and these statistics show that there are more events ending at the dawn side. We shall add this figure and corresponding text into the revised version. For MMS statistics this is probably a bit more difficult, as it requires a database of jets for Qperp and Qpar bow shock, hence we leave it as a topic of a follow-up paper.

*"My third scientific request is that the authors tell what the vertical pressure stripes are in the magnetosheath. They are so prominent in Figure 1. What is causing them?"*

The large-scale flow pattern in the magnetosheath in Vlasiator is as expected: streamlines diverge from the Sun-Earth line. However, the distribution of the velocity magnitude and the density is more complex, due to kinetic processes arising at the quasi-parallel shock, which results in the large-scale structures we observe in the magnetosheath. In essence, these stripes are remnants of the ULF wave fluctuations at the bow shock. We will add this information to the revision.

*Line 6, page 1. Using a statistics –> using statistics Line 21, Page 5. At force –> in force The stars in Figure 1 are too small. Lines 5-7 on Page 10. Can the authors state a typical dimension at this point? Line 22 on Page 10. No need for 'RE' Lines 25-26. Page 10. Can the authors state that the jets appear to be disintegrating or dying by diffusion here?*

Thank you for these comments, we shall correct them in the revision. We think based on Figure 10 that the jet "dies" through diffusion, because towards the end of its lifetime there are no steep gradients within the jet parameters (without counting the final seconds).

On behalf of all the co-authors, Minna Palmroth

---

## Author Comment (AC2)

Dear Referee #2,

Thank you for your thorough review of our paper. We have addressed all of your very good points which we think will significantly improve the quality of the paper. Below, we go through the points in detail.

*There are two major limitations of these simulations: the kinetic treatment is truncated (the kinetic physics of electrons is disregarded) and the plasma macroscopic dynamics is reduced to two spatial dimensions, in the equatorial plane. Can the authors discuss these limitations and demonstrate that the numerical simulations based on such a truncated description of plasma dynamics still captures accurately the physics of jets as observed in the magnetosheath (see, e.g., the analysis of the jet anatomy by Karlsson et al, 2018)?*

Thank you very much for this clarification.

The electron kinetic physics can be neglected for magnetosheath jets. The short answer is that the kinetic pressure of the electrons downstream of the Earth's bow shock is smaller by about a factor of 10 with respect to the ion pressure. The reason for this is that the Mach number is not large enough for the dissipation at the Earth's bow shock to necessitate heating the electrons, and thus the ions are the primary beneficiary of the heating. Since energy has to be transferred from somewhere (ion or electron thermal pressure, or magnetic pressure) to enhance the kinetic pressure (or the ram pressure) and lead to jets' formation, the primary source for such energy transfer has to be comparable in size to jets kinetic pressure. Additionally, the fact that jets have scales/dimensions between fluid and ion kinetic scales is also an indication that an ion kinetic model should be sufficient. While electrons might have some nonzero contribution in the formation of jets, the dominant effect comes from the ions. Now of course, the electron physics inside jets is important to maintain quasi-neutrality and one will certainly find electron kinetic fluctuations that are relevant to the local thermodynamical properties of the plasma, but that is a separate problem.

Regarding the 2D spatial dimensionality: The largest caveat of this is the position of the magnetopause, as we already state in the manuscript. The bow shock – magnetosheath interactions are rather accurately reproduced by 2D kinetic simulations as can be seen from a large number of previous papers (e.g., Karimabadi et al., 2014 PoP http://dx.doi.org/10.1063/1.4882875; Blanco-Cano et al al, 2006 JGR doi:10.1029/2005JA011421). The first Referee suggested a new plot looking at the final position of the jet within the dawn and dusk in the 30-degree cone angle runs. Of course the final position of the jet within the magnetosheath will lack the third dimension in a 2D simulation. However, it is important to investigate the dawn/dusk asymmetry of the final position, as the first Referee suggests.

We will add a summary of the caveats in the Discussion as requested by the Reviewer.

*The usefulness of the comparison between a statistical analysis of simulations and MMS observations of jets (collected over almost four years) is not clear to me. The authors state that MMS data serve to "verify the simulation behavior", thus, they aim to validate the hybrid Vlasov simulations with MMS data. However, there is no indication on the upstream solar wind conditions for the 6142 jets detected by MMS1. On the other hand, the statistical ensemble of 924 jets observed in hybrid Vlasov simulations is generated by only four sets of solar wind parameters (magnetic field, density, bulk velocity, cone angle and Mach number), as shown in Table 1. Therefore I am wondering how could one relate, compare and validate simulations of the magnetosheath jets as resulting from these four sets of solar wind parameters with the observations of magnetosheath jets corresponding to 6142 MMS events, and therefore for a much larger set of solar wind conditions ? The authors do show that MMS observations for one jet (figure 2) correspond to a solar wind*

*velocity "around 410 kms−1, density about 3.8 particles per cubic centimetre, the magnetic field vector about [−4, 3, −2] nT, and MA is 7" relatively close to the initial conditions assumed for the simulations reported in case LM30. However, even for this case there are significant differences in terms of jet characteristics, as outlined by the authors: "The velocity and the magnetic field show some discrepancies, such as slower flows and more variable magnetic fields at MMS, reflecting the differences in the solar wind conditions as well as the different relative positions within the magnetosheath. [. . .] Similarly, the differences in the density and temperature can be understood in terms of the differences in the solar wind parameters". Although in Figure 5 is suggested a normalization with respect to solar wind parameters, it is not clear where the solar wind parameters are derived from in case of MMS data.*

Thank you for this important clarification. The statistical comparison can never be made in an exact manner for the following reason: With an infinite amount of computational resources we could run the 6142 events, detect jets from all these runs, and then make the comparison. However, even in this case, there would be discrepancies, because in the observational data set one gets one event per solar wind condition, while in the simulation one set of solar wind conditions will produce a large amount of jets. The comparison would be accurate only if the observations would gather jets simultaneously from all around the magnetosheath, which is as impossible with current resources as it is to run 6142 kinetic simulation runs.

We would like to emphasise that the main comparison is being done through variables that have subtracted the background magnetosheath measurements ($\Delta X$). Due to the differences in solar wind conditions, we are also using normalization to the solar wind values as units (Figures 5,6). This indirectly solves the discussed issue. The plasma behind the bow shock regardless of the phenomenon we study is of solar wind origin. While variation in the observations will certainly depend on the solar wind conditions, by subtracting the background conditions and normalizing to the solar wind we are minimizing this effect.

We would also like to mention that the larger number of observational jets is very important to statistically verify the Vlasiator results. This is because then we avoid cherry-picking the data observed by MMS, while we are not in control of its orbit or position.

*The supplementary video material raises additional questions. The magnetosheath can be roughly identified in this movie as the spatial region between (an imaginary) magnetopause (as in Figure 3 of the manuscript – the Shue model) and the red line (the bow-shock). (The authors point out the magnetopause cannot be easily identified in their simulations). It looks like the dynamical pressure irregularities do not follow the general streaming or the magnetosheath flow as they seem to be predominantly convected in the duskward direction, which is puzzling. From the general gasdynamics of a supersonic flow around an obstacle one would expect a stagnation point in the nose region and azymuthal flows, duskward and dawnward, at the two flanks.*

This is an excellent question. The large-scale flow pattern in the magnetosheath in Vlasiator is as expected: streamlines diverge from the Sun-Earth line. The structures follow the field lines as they pile up and drape around the magnetopause. However, the distribution of the velocity magnitude and the density is more complex, due to kinetic processes arising at the quasi-parallel shock, which results in the large-scale structures we observe in the magnetosheath. In essence, these stripes are remnants of the ULF wave fluctuations at the bow shock, and they can only become evident in a simulation that reproduces bow shock – magnetosheath interactions, i.e., they are not visible in a fluid description. We will add this information to the revision.

*Also, one notes that some jets (identified by red dots) apparently occur spontaneously at significant distances from the bow-shock, in the magnetosheath, where they spend some time and then disappear ("die") (e.g. the sequence between t=551 and t=557 seconds, roughly at X=12, Y=1). I am not sure*

*this behavior really captures the dynamics of magnetosheath jets which are believed to be structures born at the bow-shock (e.g., Karlsson et al., 2018). When watching the movie in the supplementary material I am wondering if some of the structures considered in this study are not in fact local magnetosheath fluctuations (some of them related perhaps to the inherent numerical simulation noise and/or truncated physics) and not jets, as they are defined in the literature.*

We did in fact discuss before submitting, whether we should restrict to events which are born at the bow shock, or just take all events which fulfil the criterion. We decided to take all events fulfilling the criterion because in the observations one cannot do this restriction: No observational jet statistics can restrict to cases originating in the bow shock because they have no idea whether the event they observe in fact did originate there, or whether the magnetosheath just altered the flow conditions such that the local parameters fulfil the jet criteria. We do suspect that this occurs in many statistical data sets, and this is also why we limit to smaller jets to avoid large regions near flanks (page 5, lines 21-23 of the submitted version). We actually have a follow-up paper to be submitted in 2020, where we restrict to cases originating at the bow shock. We will add this information to the revision.

Further, we would like to add that the origin of jets is still controversial (see the review by Plaschke et al., 2018 https://doi.org/10.1007/s11214-018-0516-3). Some of the generation mechanisms do not involve bow shock processes, as e.g., some authors have proposed that magnetosheath reconnection could produce jets (Retino et al, 2007 https://doi.org/10.1038/nphys574).

*The statistical analysis of simulations results is interesting. However, given the complexity of this type of numerical simulations, the authors should give more evidence on how the results depend on the numerical setup and noise. I assume such tests were extensively performed and the authors should discuss if and how the results depend on the spatial and velocity resolution, as well as the time step assumed for the simulations. This is important for understanding the results of the statistical analysis.*

The spatial resolution is better in Vlasiator than in any of the existing ion-kinetic simulation studying the jets currently. The same goes for the velocity space resolution. Timestep is determined by the CFL conditions, which it fulfils, and therefore we do not think that the timestep is an issue. We would like to emphasise that Vlasiator is a supercomputer simulation, requiring about 3-5 million core-hours per run, and around 10 T of disk space to save the results. Unfortunately, we cannot afford large-scale numerical tests with Vlasiator. However, we have carefully analysed the minimum requirements for spatial and velocity space resolution with a simpler setup of Vlasiator solvers (Pfau-Kempf et al., 2018, doi:10.3389/fphy.2018.00044), and we have carefully chosen all parameters so that the physics of jets is properly described. Indeed, we can miss the smallest jets due to our resolution, but such small jets would also be missed from spacecraft observations. We shall add this information in the revision.

*The jet size distribution, as revealed by Figures 7 and 8, indicates that the structures studied with the hybrid Vlasov simulations are rather small scale and short lived, with most probable sizes of the order of hundred of kilometers and most probable lifetimes of the order of seconds. While lifetimes were not yet investigated with in-situ data, scale sizes derived from observations (e.g. Plaschke et al, 2016) are orders of magnitude larger, as noted by the authors themselves. This difference leads me back to the remark I made in the previous paragraph: are the structures studied in this report magnetosheath jets or local non-homogeneities downstream the bow-shock, and satisfying the selection criteria based on the dynamic pressure excess (Eq. 1) ?*

The answer is shortly: Yes. The previous literature of magnetosheath jets based on observational statistics has no way of confirming that the features they observe are in fact born at the bow shock. The only way to characterise the jets is to define a set of criteria and follow them; this is what we have done, and as our results are so closely in agreement with the observations, we can say that both

jets born at the bow shock (which only Vlasiator can confirm) and the local homogeneities fulfilling the criteria are in accordance with observations. Both types are called magnetosheath jets in observational literature, while only models can state something of their origin.

We would also like to point out that the size of the jets is a function of the criteria used. In our previous paper (Palmroth et al., 2018, reference in the manuscript), we studied the Vlasiator jet with the Plaschke criterion along with the Archer and Horbury criterion we have in the present paper. The Plaschke criterion produces larger jets than the Archer and Horbury criterion, and hence our results on the size and dimension are not directly comparable to Plaschke et al 2016 statistics. Regarding Plaschke criterion, a new paper by them (https://agupubs.onlinelibrary.wiley.com/doi/abs/10.1029/2020JA027962) states in its keypoints that "*Most magnetosheath jets are an order of magnitude smaller than previously reported*", and in Discussion they state "*First, jets are, in general, much smaller than reported in most, if not all, literature on this subject so far, where scale sizes on the order of 1RE are typically stated. Median scale sizes of D,perp = 0.12RE and D,par = 0.15RE*". These results agree with our results. Finally, we stress here that there is also a bias in observations towards larger structures: smaller jets won't be detected from spacecraft measurements.

We shall add the above information to the revision.

*Finally, I have a question for the authors. This manuscript, and many others dedicated to magnetosheath jets, presents statistics of various parameters of jets in order to describe their properties. Thus, it is implicitly recognized that these jets are plasma structures to which we can assign a shape, a size, a bulk velocity, a temperature, a lifetime. My question for the authors is which is, in their opinion, the physical process that grants the jets their individual identity, allowing them to have an individual dynamics, apparently different/independent than/from the background magnetosheath.*

Thank you for this enjoyable, a bit more philosophical question, if we may use this characteristic for the Reviewer's words. As we can see from Figures 8 and 9, the jets constitute a region of plasma that has very different properties than the surrounding magnetosheath. It is perhaps like the injection of a bubble of cold air into hotter air, or a low-pressure weather system. Of course it will mix with the surroundings eventually, but it can have a clear identity. We could also compare jets to bursty bulk flows in the magnetospheric tail. While BBFs are considered well-identified entities based only on their velocity difference from the background, jets have, apart from the velocity, also other properties different than the background, such as temperature.

*I see an issue with the availability of simulations and MMS data on which the findings of this manuscript are based. I am wondering if there is compliance with Annales Geophsycae policy on data availability. Indeed, the authors indicate Acknowledgments that "Data presented in this paper can be accessed by following the data policy on the Vlasiator web site." On the Vlasiator website I found the following statement regarding simulations data availability: "All data requests and other support questions should be addressed to the PI. The PI-team decides about the time and place in which the peer-reviewed data becomes public." I think the authors should demonstrate the simulations and experimental data are available from a public data repository as requested by Annales Geophysicae data policy (https://www.annales- geophysicae.net/about/data_policy.html), I quote : "It is particularly important that data and other information underpinning the research findings are "findable, accessible, interoperable, and reusable" (FAIR) not only for humans but also for machines. There- fore, Copernicus Publications requests depositing data that correspond to journal articles in reliable (public) data repositories, assigning digital object identifiers, and properly citing data sets as individual contributions. [...] A data citation in a publication resembles a bibliographic citation and needs to be included in the publication's reference list. [...] In addition,*

*data sets, model code, video supplements, video abstracts, International Geo Sample Numbers, and other digital assets should be linked to the article through DOIs in the assets tab."*

The Vlasiator simulation itself is open source and freely executable by anyone wishing to reproduce these results. One needs to download the source code from the Git repository, follow the guidelines given in the manuscript to make the runs, and then do the postprocessing as explained in the manuscript. The request of the data from the PI works such that a person indicates which data they are interested of, and we will then be able to give that exact data. Our analysis software is open source, and hence the given data can again be post-processed by the requesting side openly and freely. Therefore, we do fulfil the FAIR principles. The reason we do not give all run data openly on a web service is because the size of our run data base so far is well above 200 T. There is no such system currently that would make it possible for us to give these data (and metadata), and to make this kind of a system from scratch would require a huge amount of coding, which is outside of the scientific work required by the funding agencies which fund our work.

*Minor remark: Section 2.2. consists on a single paragraph. I suggest the authors expand it and include a description of the MMS data used in their study or integrate the paragraph in the previous section.*

We shall revise this in the revision.

On behalf of all the co-authors,
Minna Palmroth

---

## Author Response (AR2)

Espoo, 1.2.2021

Dear Referee #2,

Thank you for your thorough review of our paper. We have addressed all of your very good points which we think will significantly improve the quality of the paper. Below, we go through the points in detail.

*The authors should clarify their claim that jets can be spontaneously born in the magnetosheath and reach levels of dynamic pressure larger than the background magnetosheath plasma state. Which physical process would allow for such extreme nonlinear plasma behavior? Also, the authors conjecture that some experimental observations of jets might be „contaminated" by spontaneously born structures like the ones observed in their simulations, looks too strong.*

Thank you for this comment. There is a misunderstanding. We do not claim that jets can be spontaneously born within the magnetosheath. We mean that within the magnetosheath different processes can change the overall conditions such that local regions within the magnetosheath fulfill the jet criteria. These have been mentioned e.g., by Hao et al. (https://doi.org/10.1002/2015JA021419). These regions are obviously not those jets that the literature speaks of, i.e., those which are thought to originate at the bow shock due to bow shock – foreshock interactions (see e.g., the review by Plaschke et al., 2018, reference in the manuscript). The processes which can make the local conditions to fulfill the jet criteria are, for example, magnetosheath waves, and the transferred ULF waves from the foreshock (which can be seen as pressure ridges within the magnetosheath e.g., in Fig. 1) Since the spacecraft has no way of knowing whether a detected pressure increase formed locally or whether it was carried by the flow from the bow shock, these local regions cannot be removed from the observational jet statistics. Therefore we do not take them away from our statistics in this paper, to facilitate comparison with spacecraft observations. We have tried to make this point clearer in the paper.

*The argument put forward to explain why a truncated approach disregarding electron kinetics is valid, is not convincing : „As for electrons, the electron kinetic physics can be neglected for magnetosheath jets mainly because the kinetic pressure of the electrons downstream of the Earth's bow shock is smaller by about a factor of 10 with respect to the ion pressure. Additionally, the fact that the jet dimensions are between fluid and ion kinetic scales is also an indication that an ion kinetic model should be sufficient. Naturally the electron physics inside jets is important to maintain quasi-neutrality and one will certainly find electron kinetic fluctuations that are relevant to the local thermodynamical properties of the plasma, but that is a separate problem. Hence, we conclude that the 2D3V approach neglecting kinetic electrons is sufficient to investigate the jet formation and transfer through the magnetosheath." I understand the discussion on the electron to ion kinetic pressure ratio, however, the environment simulated numerically is a collisionless magnetized plasma where the motion of particles is dominated by the magnetic and electric fields. It was shown that electron dynamics at kinetic scales, i.e. the one where the spatial scales of the order of electron Larmor radius is resolved, play a crucial role for the electric and magnetic configuration (see classical plasma textbooks like, e.g., G. Schmidt, 1966). Obviously, I do not ask the authors to perform global fully kinetic Vlasov simulations, I believe they could use some insight from theory and local 3D fully kinetic particle in cell simulations to put their simulations in the right context.*

Thank you for directing us to the Schmidt book. We have now spent time in studying 3D fully kinetic simulation papers to investigate the role of electron dynamics in the propagation of the jets. One particularly interesting paper, Voitcu and Echim (JGR 2016, doi:10.1002/2015JA021973) investigates idealized plasma bubbles propagating in transverse magnetic fields mimicking

northward IMF. They find that a polarization electric field forms inside the "jet" (i.e. a bubble), and it contributes to the propagation of the "jet" across the transverse magnetic field. This process is of course neglected in our simulation. What is a pity is that the Voitcu and Echim (JGR2016) limit their studies to the transverse IMF, making it difficult to assess how the polarization electric field would affect our simulations. This is because we have a radial to Parker spiral IMF. In the radial runs the field is highly turbulent at the subsolar magnetosheath, and in the 30° cone angle runs a bit less turbulent but definitely not full-on transverse. Since according to the Voitcu and Echim papers the polarization electric field contributes to the forward propagation of the jets under transverse fields, we are not entirely sure how the effect would affect our jet propagation results; maybe primarily to the speed of the jets, which is not under scrutiny in the paper. However, as this has not been studied, we leave the more direct speculation out from the paper and note that this should be investigated in future papers. We added discussion about this effect to the paper.

*A note on the normalization used in figure 5: all the physical parameters are normalized to solar wind conditions. As I asked in my previous report, to which exactly values of the solar wind variables ? Insofar simulation results are concerned, this is rather clear. I assume the normalization is performed with respect to the solar wind parameters given in Table 1. But what about normalization of MMS data? I understand the solar wind data are taken from OMNI but which values, at which moment of time ? To be more explicit: there are 6142 jets in MMS data base. I assume one value per jet is included in the histograms shown in the 3rd column of figures 5 and 6. The question is which value of the OMNI solar wind density, velocity, dynamic pressure, magnetic field, temperature is used to normalize each of the MMS data shown in Figure 5 and 6. Is this an OMNI sample extracted at exactly same UT as the MMS sample ? Or the normalizing value is derived from an average over a time interval ? Do the authors consider some time delay between OMNI and MMS data ? This question seems important to me given the variability of solar wind parameters, thus choosing one or another solar wind sample might significantly change the shape of MMS distributions shown in figures 5 and 6.*

First, similar association of OMNI values has been done in [Raptis et al., 2020a, Raptis et al., 2020b] using the same MMS dataset of jets. Furthermore, other works using similar datasets also followed a similar approach (e.g. [Vuorinen et al., 2019, Plaschke et al., 2020] using THEMIS data). Finally, in [Raptis et al., 2020b] there is an extended analysis on what can be done for the association to be as accurate as possible while discussing possible limitations. For the current paper we did something similar for the normalization and for the OMNIweb association to ensure an accurate comparison. Specifically,

1. As mentioned in the current version of the manuscript (p.8 , lines 6-7) we remove jets that have a maximum standard deviation of the magnetic field rotation angle higher than $45°$

2. OMNI data were (a) time shifted/lagged and (b) averaged for an ideal match to the jets observation at the magnetosheath.

This was done by taking an average of 20 (1-min resolution) data points from OMNI, starting 15 minutes before the jet observation time and up to 5 minutes after the jet observation time.

It should be noted that the OMNIweb data we are referring are the 1-min high resolution, propagated to the bow shock nose data, taken from https://omniweb.gsfc.nasa.gov/ow_min.html

This unequal averaging was done because the average time from the bow shock to MMS location is ∼ 5 minutes as discussed in [Raptis et al., 2020b]. As a result, by taking under consideration this

time lag effect (5 min), we effectively take a ± 10 minutes window around the jet's associated solar wind measurements.

Therefore, we not only remove the extremely varying solar wind conditions (1) but we also take care of (2a) the time shift required from the bow shock to magnetosheath and (2b) possible variations/errors of the OMNI database by taking a 20 minute average for the conditions. We added this information to the manuscript.

*I have a remark on authors claim: "Figure 5 shows an excellent overall agreement especially between the Vlasiator jets at random times and the MMS jets." When one looks at the statistical results, the overall agreement is not so excellent. Indeed, Figure 5 shows significant differences between the numerical simulations and MMS histograms/distributions: (1) extend of jets: the standard deviation of MMS observations is one order of magnitude larger than simulations meaning that observed jets span a much broader range of scales; (2) density of jets: the MMS distribution is skewed towards larger (normalized) values while simulations distribution is more Gaussian meaning that MMS observations indicate a preference towards larger (normalized) densities; (3) maximum value of velocity: the skewness of numerical simulations and MMS are significantly different, indicating different probabilities in numerical versus MMS data for different ranges of velocities; (4) maximum of the dynamic pressure: numerical simulations show a flat top distribution while the MMS distribution is skewed with one peak.*

Regarding Figure 5, we think it is to a large extent a matter of taste what one means by "excellent agreement". We mean this from the perspective that we have four runs spanning four solar wind conditions, and have a finite maximum resolution in the simulation restricting to investigate the scale separation. We have reformulated the wording.

*I note the authors conjecture: „As we can see from Figures 8 and 9, the jets constitute a region of plasma that has very different properties than the surrounding magnetosheath. It is perhaps like the injection of a bubble of cold air into hotter air, or a low-pressure weather system". I tend to believe they are right. Nevertheless, this brings back my criticism about neglecting the kinetic physics of electrons. Indeed, earlier theoretical papers and recent local three-dimensional, fully electromagnetic kinetic (particle in cell) simulations of jets (also called plasmoids in some earlier publications) indicate that the edges of the „bubble" are precisely the key sites where electron kinetic physics is important and effective on the dynamics of the jet, regardless the electron to ion pressure ratio.*

Thank you for this comment. We have formulated the discussion on electrons referring to studies by Voitcu and Echim (2016, 2018) which address the role of electron dynamics, however, unfortunately this cannot be studied in this paper. We have recently developed an electron capability within Vlasiator (Battarbee et al., https://doi.org/10.5194/angeo-2020-31), which we could use in the future to see this in more detail.

On behalf of all the co-authors,
Minna Palmroth

References

[Plaschke et al., 2020] Plaschke, F., Hietala, H., and Vörös, Z. (2020). Scale sizes of magnetosheath jets. *Journal of Geophysical Research: Space Physics*, 125(9):e2020JA027962. e2020JA027962 10.1029/2020JA027962.

[Raptis et al., 2020a] Raptis, S., Aminalragia-Giamini, S., Karlsson, T., and Lindberg, M. (2020a). Classi- fication of magnetosheath jets using neural networks and high resolution OMNI (HRO) data. *Frontiers in Astronomy and Space Sciences*, 7:24.

[Raptis et al., 2020b] Raptis, S., Karlsson, T., Plaschke, F., Kullen, A., and Lindqvist, P.-A. (2020b). Clas- sifying magnetosheath jets using MMS: Statistical properties. *Journal of Geophysical Research: Space Physics*, 125(11):e2019JA027754. e2019JA027754 10.1029/2019JA027754.

[Vuorinen et al., 2019] Vuorinen, L., Hietala, H., and Plaschke, F. (2019). Jets in the magnetosheath: IMF control of where they occur. In *Annales Geophysicae*, volume 37, pages 689–697. Copernicus GmbH.